



**Spatial–Seasonal Isotopic Variations in a Surface–Groundwater System in an**
**Extremely Arid Basin and the Associated Hydrogeological Indications**
Yu Zhang[1], Hongbing Tan[1, *], Peixin Cong[1], Dongping Shi[1], Wenbo Rao[1], Xiying Zhang[2]
[1]School of Earth Sciences and Engineering, Hohai University, Nanjing 210098, China
[2]Qinghai Institute of Salt Lakes, CAS, Xining 810008, China
**\* Corresponding author:** Hongbing Tan (tan815@sina.com)
**Abstract**
Climate warming accelerates the global water cycle. However, the relationships between
climate warming and hydrological processes in the alpine arid regions remain unclear. Herein,
high spatiotemporal resolution sampling of surface water and groundwater was performed at the
Qaidam Basin, an extremely arid area in the northeastern Tibetan Plateau. Stable H-O isotopes and
radioactive [3]H isotopes were combined with atmospheric simulations to examine climate change
and hydrogeological characteristics. The surface water heavy isotopes enrich during the wet season
and deplete during the dry season. The contribution of precipitation to river discharge was
considerably higher in the eastern region of the basin (approximately 45%) than in the central and
western regions (10%−15%). The H-O isotopic compositions showed a gradually negative spatial
pattern from the west to the east in the Eastern Kunlun Mountains water system; a reverse pattern
occurred in the Qilian Mountains water system. This distribution pattern was jointly regulated by
the westerly water vapor transport intensity and local hydrothermal conditions. Increased
precipitation and cryosphere shrinkage caused by climate warming mainly accelerated basin
groundwater cycle. In the eastern and southwestern Qaidam Basin, precipitation and ice/snow
meltwater infiltrate structural channels that favor water flow, such as fractures and fissures,
facilitating rapid seasonal groundwater recharge and increased terrestrial water storage. However,
under future increases in precipitation in the southwestern Qaidam Basin, compensating for water
loss from long-term melting of ice and snow will be challenging, and the total water resources may
show an initially increasing and then decreasing trend.
**Keywords:** Qaidam Basin; isotope hydrology; water cycle; spatiotemporal pattern; climate





## 1. Introduction

Amidst the impending climate change process, an in-depth study of the hydrological cycle processes is a prerequisite for implementing water resource management and trend forecasting. Over the past half-century, continuous climate warming and intensified human activities have led to global water cycle acceleration and water resource redistribution at different scales (Huntington et al., 2006; Durack et al., 2012; Masson-Delmotte et al., 2021). For example, rapid warming has driven the rapid expansion of lakes in the Tibetan Plateau and the shrinking of lakes in the Mongolian Plateau (Zhang et al., 2017), and it has also amplified the severe shortage of irrigation water in parts of South Asia and East Asia (Haddeland et al., 2014). Moreover, it is expected to also reduce groundwater storage in the western region of the United States (Condon et al., 2020). At present, the arid regions of northwestern China are undergoing a change in climate from warm–dry to warm–humid (Zhang et al., 2021). The resulting uncertainties in alpine arid basin water resources in this region present new challenges in understanding the hydrological cycle and present state of water resources. These key scientific issues can be resolved by investigating the spatiotemporal distribution and driving mechanisms of surface and groundwater resources in the basin under accelerating climate warming.

The Tibetan Plateau, also known as the "Third Pole", has complex cryosphere-hydrology-geodynamic processes and is especially susceptible to global warming (Zhang et al., 2017; Yao et al., 2022). The Qaidam Basin is situated in the northeastern Tibetan Plateau and presents the largest extent of warming in the entire Tibetan Plateau and a substantial steep rise in temperature globally (Li et al., 2015; Kuang and Jiao, 2016; Yao et al., 2022). Since 1961, the average temperature of the basin has been rising at an alarming rate of 0.53°C/10 a as a result of climate warming (Wang et al., 2014), resulting in an increase in precipitation and retreat of the cryosphere in the region (Song et al., 2014; Xiang et al., 2016; Zou et al., 2022). These changes have led to rapid spatial changes in water storage in the Qaidam Basin, increased runoff or rising groundwater table in most parts of the region (Jiao et al., 2015; Wei et al., 2021), and hydrological changes, such as the expansion of lakes, in the central and northern regions of the basin (Ke et al., 2022; Zhang et al., 2022). However, several questions remain to be resolved: How are hydrological changes in the basin driven by climate changes? What are the potential influences of these changes on the water resources of the basin? These issues require an in-depth investigation. Rivers and groundwater carry precipitation and meltwater from high-altitude areas to the lakes in low-lying areas;



information on climate-hydrology dynamics of the runoff process can provide key evidence for
the entire water cycle process. Hence, the Qaidam Basin is an excellent site for investigating the
response mechanism of the hydrological cycle in the Tibetan Plateau caused by global warming.
Water isotopes (H and O) represent important components of water molecules and are useful
natural environmental tracers of the water cycle and climate reconstruction, and they can help
elucidate the processes that control water cycle changes, thus providing scientific evidence for
human adaptations and effects on future global changes (Craig, 1961; Dansgaard, 1964; Yao et al.,
2013; Bowen et al., 2019; Kong et al., 2019). Stable water isotope records provide key information
on water migration processes, and they can compensate for the paucity of hydrometeorological,
geological, and borehole data in hydrological research. Stable H-O isotopes and radioactive $^{3}$H
isotopes have been widely applied to quantify surface or groundwater recharge sources,
interactions, budgets, and ages (Befus et al., 2017; Moran et al., 2019; Bam et al., 2020; Shi et al.,
2021; Ahmed et al., 2022). Previous researchers have also performed a substantial amount of work
on the use of isotopes to trace the water cycle in the Qaidam Basin (Xu et al., 2017; Xiao et al.,
2017, 2018; Zhao et al., 2018; Tan et al., 2021; Yang and Wang, 2020; Yang et al., 2021). These
studies have enhanced our understanding of aquifer properties in local regions and their recharge
mechanisms. However, under the constraints of the harsh climate environment and
hydrogeological survey accessibility in the Qaidam Basin, the water cycle processes in the basin
in previous reports are mainly understood based on the watershed or confined regional unit scale.
The use of regional research to achieve a comprehensive elucidation of the basin-scale water cycle
mechanism is a challenge. Furthermore, the surface water and groundwater seasonal recharge
information of the whole basin has not been systematically explored. Continuous changes in the
topographical and tectonic spatial patterns of the basin are caused by various hydrological, climatic,
and hydrogeological conditions; moreover, the hydrological effects exerted by anthropogenic
climate change and regional aquifer properties differ seasonally (Jasechko et al., 2014). Therefore,
it is particularly essential to study the entire process of basin water cycle and seasonal changes.
While carrying out a comprehensive assessment of differences in isotopes of various potential
recharge sources, it is fundamental to use the same technical methods for the systematic sampling
and isotopic characterization of the basin.
In this study, the eight major watersheds of the Qaidam Basin were selected as the study sites
and constraints were placed on the hydrological cycle patterns and processes of the Qaidam Basin





based on stable H-O isotope and radioactive $^3$H isotope data from the wet–dry season. The aims
were to 1) elucidate the spatial–seasonal distribution pattern of surface–groundwater isotopes at
different watershed scales in alpine arid basin; 2) analyze the composition changes of the Qaidam
Basin water sources at different spatial–seasonal scales; 3) trace the entire water cycle process
around the mountain–basin watersheds of the Qaidam Basin; and 4) predict the trend in the changes
of Qaidam Basin water resources under the influence of a large extent of climate warming. The
scientific contributions of this study include clarifying the isotope hydrology responses to climate
change in the Tibetan Plateau arid basin, which is one of the ecosystems most affected by climate
warming worldwide, from a microscopic scale; predicting the changing trend of water resources
under the condition of multiple water sources recharge; and elucidating the entire water cycle
process in extremely arid basins under the influence of rapid climate change.
**2. Study area**
2.1. General features
The Qaidam Basin is a closed and huge fault basin situated in the northeastern Tibetan Plateau
(Figures 1a and 1b). With an area of approximately 250,000 km$^2$, the basin is one of the four main
basins in China, and it is surrounded by the Kunlun Mountains, Qilian Mountains and Altun
Mountains. The Qaidam Basin has a plateau continental climate and represents a typical alpine
arid inland basin that is characterized by drought. There are substantial variations in the basin
temperature, and the average annual temperature is below 5 °C. The annual precipitation declines
from 200 mm in the southeastern region to 15 mm in the northwestern region. The average annual
relative humidity is 30%–40%, with a minimum lower than 5%. Modern glaciers have formed in
the mountains on the southern and northern sides of the basin, which is surrounded more than 100
rivers. Approximately 10 rivers are permanent, with most of the local rivers representing
intermittent river systems. The rivers are mainly distributed on the eastern side of the basin but
scant on the western side. The water in the basin lakes has become predominantly saline,
comprising 31 salt lakes in total.



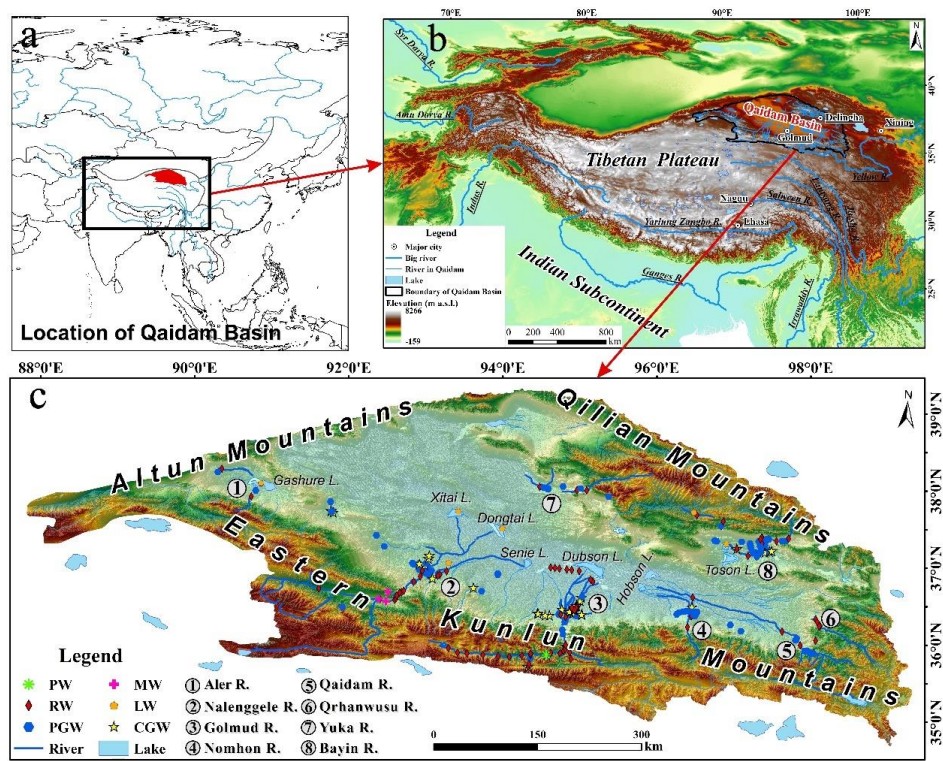

**Figure 1.** Location of Qaidam Basin (a, b) and the sampling sites (c)

2.2 Hydrogeology and structure

The basin basement consists of Precambrian crystalline metamorphic rock series, and the caprock is of Paleogene-Neogene and Quaternary strata. The mountainous area surrounding the basin is dominated by a Paleogene system, and the basin area and basin boundary zone are characterized by a wide distribution of the Paleogene-Neogene system. The Quaternary system is mainly distributed in the central basin region and the intermountain valley region. The basin terrain is slightly tilted from the northwest to southeast, and the height gradually reduces from 3000 m to approximately 2600 m. The distribution of the basin landforms presents a concentric ring shape. From the edge to the center, the distribution of diluvial gravel fan-shaped land (Gobi), alluvial–diluvial silt plain, lacustrine–alluvial silt clay plain, and lacustrine silt–salt plains follow a regular pattern. Salt lakes are extensively distributed in low-lying terrains. The inner edge of the Gobi belt in the northwestern basin region is clustered with hills that are less than 100 m in height. The





southeastern region of the basin has marked subsidence, and the alluvial and lacustrine plains are
expansive. In the northeastern basin region, a secondary small intermountain basin has been
formed between the basin and the Qilian Mountains by the uplifting of a series of low mountain
fault blocks of metamorphic rock series.
The Qaidam Basin is situated in the Qin-Qi-Kun tectonic system, where there is strong
neotectonic movement, and a series of syncline-anticline tectonic belts and regional deep faults
have formed around it. The fault structures in the Qaidam Basin are very well developed and
include the north-easterly Alun fault in the north; north-westerly Saishenteng–Aimunik northern
margin deep fault in the northeast; westerly Qaidam northern margin deep fault in the northwest;
Qimantag Mountains and Burhan Budai Mountains–Aimunik northern margin deep fault in the
south; and north-westerly Sanhu major fault and north-easterly Qigaisu–Dongku Fault in the
central basin region.
The basin water system distribution is subject to the constraints of the topography and
neotectonic movements, and it appears to present an overall centripetal radial pattern (Figure 1c).
There is frequent surface water–groundwater exchange, which is generally manifested as an
abundance of precipitation and ice/snow meltwater in the mountainous areas, which are the main
runoff areas. The runoff from the mountain flows through the Gobi belt in front of the mountain,
with most of it infiltrating into the groundwater, subsequently flows over the surface in the form
of confined artesian water or a spring at the front edge of the alluvial fan, and finally flows into
the terminal lake. Groundwater can be roughly divided into bedrock fissure water, leached pore
water and local confined groundwater, phreatic groundwater and confined artesian water, as well
as salty phreatic groundwater, brine, and salty confined artesian water. Surface water and
groundwater salinity and solutes are gradually enriched along this process (Wang et al., 2008).



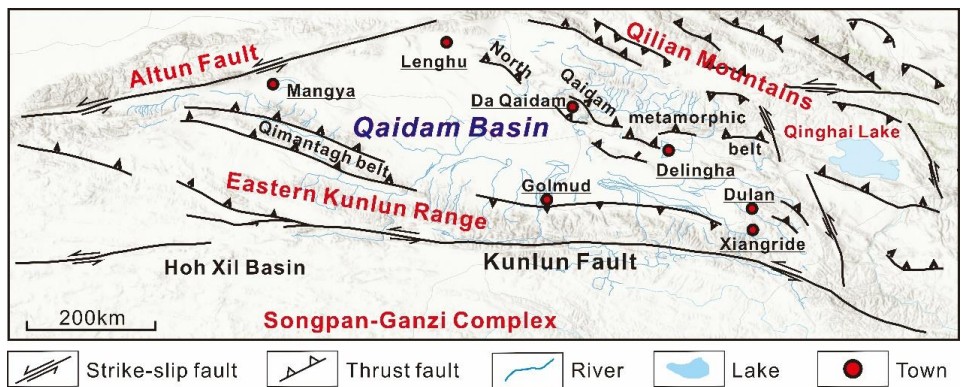


**Figure 2.** Map of the Qaidam Basin tectonic distribution (modified after Jian et al., 2020)

**3. Sampling and methods**

3.1 Sampling and analysis

To elucidate the water cycle mechanisms in the Qaidam Basin, field investigations and sampling were carried out on the 8 major river–groundwater systems in the region from 2019 to 2021. The sampling frequency of 6 watersheds essentially extends over a hydrological year and is represented by the wet season (July–August) and the dry season (March–April). Precipitation and snow meltwater were collected from the Eastern Kunlun Mountains. In total, 239 sampling points were established: phrenic groundwater (n = 100), confined groundwater (n = 43), spring water (n = 6), river water (n = 81), lake water (n = 5), snow meltwater (n = 3), and precipitation (n= 1). A total of 422 sets of samples were collected. No sampling point was established in the northwestern basin because the southern slope and front edge of the Altun Mountains consisted of tertiary system halite sedimentation and Quaternary system large salt flats, and no freshwater body is present. Therefore, the sample collection covers the entire Qaidam Basin and each of the major endorheic regions.

Hydrogen and oxygen isotope ($^2$H, $^3$H, and $^{18}$O) tests were conducted at the State Key Laboratory of Hydrology-Water Resources and Hydraulic Engineering, Hohai University, China. A MAT253 mass spectrometer was used to measure the ratios of $^2$H/$^1$H and $^{18}$O/$^{16}$O, and the results were compared with the Vienna Standard Mean Ocean Water (VSMOW), expressed in δ (‰), with the analytical precision (1σ) of the instrument for these isotopes was lower than ±1‰ and ±0.1‰. To determine the tritium ($^3$H) concentration, the water sample was first concentrated by





electrolysis. Following sample enrichment, measurements were carried out using low background

liquid scintillation counting (TRI-CARB 3170 TR/SL). The findings were expressed in terms of

absolute concentration in tritium units (TU), the detection limit of the instrument was 0.2 TU, and

the precision was improved to more than ± 0.8 TU.

3.2 Hydrograph separation

In the analysis of water sources among hydrological processes, endmember mixing models

are widely used. According to the heterogeneity of different end member isotopes/water chemistry

parameters, combined with the Bayesian mixing model, the contribution of each recharge end

member to the mixed water body can be estimated (Hooper et al., 1990, 2003; Chang et al., 2018).

The process is as follows:

$$1 = \sum_{i=1}^{n} f_i, \ \ C_m^j = \sum_{i=1}^{n} f_i C_i^j, j = 1, \dots, n \tag{1}$$

where $f_i$ represents the proportion of water source $i$, $n$ represents the number of end members, and

$C_m^j$ represents the level of tracer $j$ in end member $i$.

Stable isotope analysis in R based on Bayesian mixing models (MixSIAR) can quantify the

contributions of more than two potential endmembers (Parnell et al., 2010). In this study, based on

the differences in the water body properties and isotopic composition of each endmember, $\delta^{18}$O,

$\delta$D, and d-excess (d-excess = $\delta$D − 8$\delta^{18}$O) data were used as parameters in the modeling. The

model calculation process was carried out at a fractional increment of 1% and an uncertainty level

of 0.1%.

3.3 Water vapor trajectory

The source and transport route of moisture can be monitored based on the water vapor flux

field derived from the monthly mean ERA5 reanalysis data (0.25° × 0.25°) of the European Centre

for Medium-Range Weather Forecasts (ECMWF, https://www.ecmwf.int/) (Hersbach et al., 2019).

In this study, after taking into account that more than 70% of the precipitation in the Qaidam Basin

occurs from June to September, the monthly mean ERA5 reanalysis data in this period from 2019

to 2021 were used to analyze the water vapor transport path in and around the study area. Based

on the average altitude of >3000 m at the study site, the simulated atmospheric pressure was set to





500 hPa. The majority of the atmospheric water vapor was distributed in the range of 0–2 km
above ground, and the simulated height did not have any remarkable influence on the findings (Li
and Garzione, 2017; Yang and Wang, 2020).
**4. Results**
4.1 Stable H and O isotopes of different water bodies
The spatiotemporal changes of isotopes in the various water bodies in the entire basin are
large, the watersheds have distinct characteristics, and considerable differences exist between
surface water and groundwater. The $\delta^{18}$O and $\delta$D values extracted from the water samples of each
watershed in the study region can be classified into six categories (Figure 3): 1) precipitation; 2)
snow meltwater; 3) river water; 4) lake water; 5) phreatic groundwater; and 6) confined
groundwater.
In the Qaidam Basin, the sources of precipitation are mainly concentrated on the northern and
southern slopes of the Kunlun Mountains and Qilian Mountains, respectively. The ranges of $\delta^{18}$O
and $\delta$D of the precipitation samples from the Kunlun Mountains and Qilian Mountains (Zhu et al.,
2015) were −23.38‰ to +2.55‰ and −158.64‰ to +30.49‰, respectively. The corresponding
fitted Local meteoric water line (LMWL) equation in the Qaidam Basin was $\delta$D = 7.48$\delta^{18}$O +
11.30 ($R^2$ = 0.95, n = 74), where the slope and intercept were similar to the long-term monitoring
findings of the Qilian Mountains (Zhao et al., 2011; Gui et al., 2020; Wu et al., 2022; Yang et al.,
2023). In the Qaidam Basin, the heavy isotopes present in snow meltwater samples were
considerably depleted compared to rainwater. The $\delta^{18}$O and $\delta$D ranges were −19.30‰ to −8.27‰
and −152.02‰ to −53.52‰ respectively, and the fitting trend equation was $\delta$D = 7.78 $\delta^{18}$O + 10.85
($R^2$ = 0.83, n = 11), with the slope and intercept lying between LMWL and GMWL (Global
meteoric water line).
Among the surface water samples, the $\delta^{18}$O and $\delta$D ranges in river water were −13.51‰ to
−5.93‰ and −85.00‰ to −47.50‰ respectively, whereas those in the lake water were more
enriched at −4.10‰ to 8.84‰ and −31.05‰ to 22.07‰, respectively. The fitted trend lines of river
and lake samples were: $\delta$D = 5.97$\delta^{18}$O − 5.54 ($R^2$ = 0.85, n = 92) and $\delta$D = 4.64$\delta^{18}$O − 16.37 ($R^2$
= 0.99, n = 7), respectively, which were below both the GMWL and LMWL, indicating that the





surface water body has undergone varying extents of evaporation, with evaporation from lakes being more enhanced.

In the groundwater samples, the H-O isotopic composition range was wider and considerable differences occurred between phreatic and confined groundwater. The $\delta^{18}O$ and $\delta D$ value distribution ranges in phreatic groundwater were −12.70‰ to −5.21‰ and −87.38‰ to −42.00‰, respectively, and the fitted trend line was $\delta D = 5.73\delta^{18}O − 9.20$ ($R^2 = 0.83$, n = 185). The phreatic groundwater isotopic composition and slope of the trend line were similar to those of surface water, indicating frequent interactions between the two and substantial evaporative fractionation of some shallow groundwater. The $\delta^{18}O$ and $\delta D$ ranges in confined groundwater were relatively small and lower in comparison at −12.12‰ to −8.58‰ and −85.00‰ to −51.01‰. The linear regression relationship of the samples fitting ($\delta D = 7.84\delta^{18}O + 12.39$, $R^2 = 0.87$, n = 51) revealed that its slope and intercept were essentially consistent with those of GMWL and LMWL, which suggests the presence of a strong correlation between confined groundwater and atmospheric precipitation in different periods.

Overall, in the Qaidam Basin, the stable H-O isotopic compositions of surface water and groundwater were generally positively skewed. The isotopic composition and trend fitting characteristics both demonstrated that the water samples have undergone varying extents of evaporation during runoff, which reflects the cold and dry climate environmental characteristics of the study area.

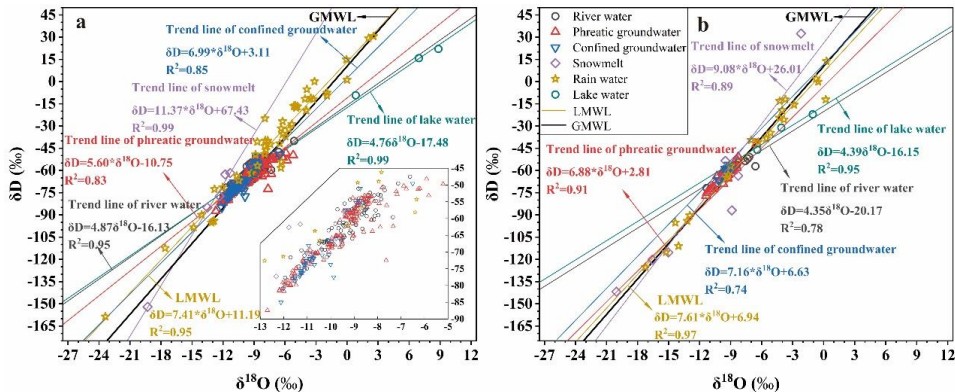

**Figure 3.** Diagram showing the relationship between $\delta^{18}O$ and $\delta D$ in different water bodies in the Qaidam Basin (a. Eastern Kunlun Mountains water system; b. Qilian Mountains water system; The data of Rain water and Snowmelt in the Qilian Mountains were from Zhu et al., 2015 and Yang et al., 2021, respectively)

### 4.2 Spatial-seasonal characteristics of surface water $\delta^{18}O$-$\delta D$

In the Qaidam Basin, considerable seasonal and spatial variations exist in the stable H-O isotopes of surface water (Figure 4). In terms of seasonal variation, apart from the Nomhon River, all watersheds displayed the characteristics of heavy isotope enrichment to varying extents during the wet season and relative depletion during the dry season. The basin surface water mean $\delta^{18}O$ and $\delta D$ values were positively skewed by −0.08‰ to 1.08‰ and 0.63‰ to 10.58‰, respectively, in the wet seasons. Moreover, the seasonal variations of $\delta^{18}O$ and $\delta D$ were more pronounced in the downstream river compared to the upstream segment. For example, the $\delta^{18}O$ value of the downstream Nomhon River was 3.66‰ higher during the wet season compared to the dry season. These phenomena reflect the differences in the recharge sources of the river during the wet–dry seasons and the strong surface evaporation effect in the central basin region. For spatial patterns, the isotopic composition of rivers originating from the Eastern Kunlun Mountains and Qilian Mountains had a contrasting distribution pattern, where the heavy isotopes of the Eastern Kunlun Mountains are gradually depleted in the direction of west to east, and the reverse held true for the Qilian Mountains. Of these, the $\delta^{18}O$ and $\delta D$ values were significantly positively skewed in the southwestern region of the basin and significantly negatively skewed in the eastern region.

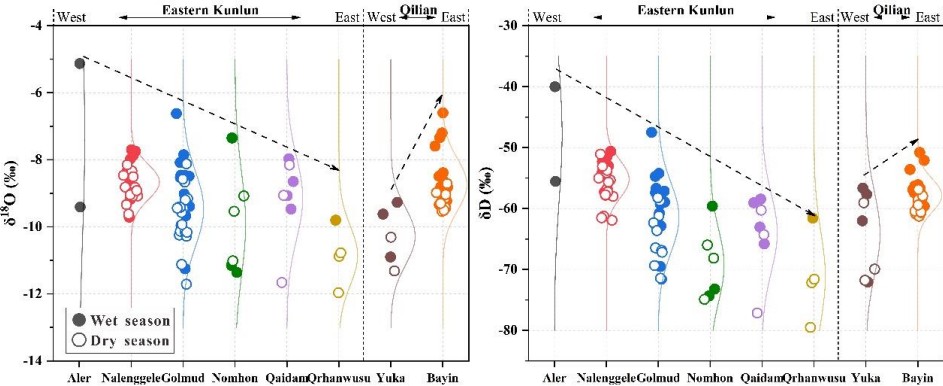

**Figure 4.** Spatiotemporal variations in the H-O isotope composition of Qaidam Basin river water

### 4.3 Spatial-seasonal characteristics of groundwater $\delta^{18}O$-$\delta D$

The spatial variability of groundwater stable H-O isotopes was more pronounced compared with river water, although it did appear to follow the same distribution pattern as river water in the watershed (Figure 5). In terms of seasons, the $\delta^{18}O$ and $\delta D$ values in the groundwater system were



lower and seasonal fluctuations were smaller compared to that of the surface water. Specifically,
the average seasonal variation of $\delta^{18}O$ in each of the groundwater systems was in the range of
$-0.75‰$ to $+0.84‰$, and the maximum seasonal variations in individual borehole were $+3.31‰$
and $-3.16‰$, respectively. This indicates that the groundwater isotopic composition was not
entirely impacted by surface water infiltration. The region with the largest seasonal fluctuations of
groundwater was located in the Nalenggele River in the southwestern basin, and the groundwater
stable H-O isotopes in wet season were significantly more positively skewed compared to those
of the dry season. Meanwhile, the region with the smallest seasonal variations of $\delta^{18}O$ and $\delta D$ is
the adjacent Golmud River. Although there were no apparent differences in the topography and
landforms of the two adjacent watersheds, significant differences were observed in the isotopic
characteristics of the two. These phenomena reflect the following: 1) the kinetic fractionation of
groundwater isotopes caused by evaporation and mixing was smaller than that of surface water
isotopes; and 2) substantial differences were detected between the groundwater recharge and
surface water–groundwater hydraulic interactions in each watershed.

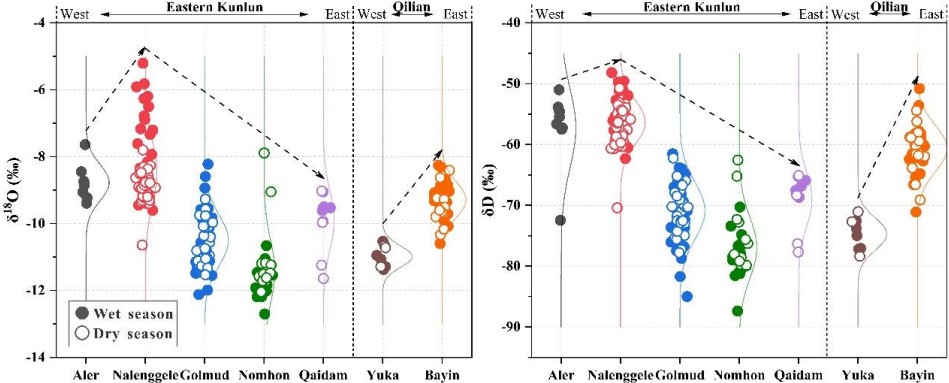

**Figure 5.** Spatiotemporal variations in H-O isotopes in the groundwater of the Qaidam Basin
**5. Discussion**
5.1 Water cycle information indicated by surface water isotopes

Owing to the scant precipitation in the alpine arid region and its concentration in summer

(June to September), the isotopic characteristics of surface water may reflect precipitation
characteristics in the respective region during the wet season. The seasonal characteristics of stable
H-O isotopes in the surface water, which consisted of enrichment during the wet season and





depletion during the dry season (Figure 4), were consistent with the observed–simulated patterns
of changes in precipitation isotopes in the basin and its surrounding areas within each year (Liu et
al., 2009; Zhao et al., 2011; Gui et al., 2020; Wu et al., 2022). In particular, the $\delta^{18}$O values at the
sampling sites in the mountainous areas on the upper stream of each watershed were positive
during the wet season compared to the dry season, which reflects the contribution of precipitation
that is enriched in heavy isotopes to the river. Moreover, the mean $\delta^{18}$O and $\delta$D values were higher
in watersheds (such as Qaidam and Bayin Rivers) during wet season, with corresponding greater
precipitation (Figure S1). From this, it can be inferred that the river water stable H-O isotopes of
each watershed in the basin were predominantly influenced by summer precipitation during the
wet season. This is largely due to the wet season coinciding with the rainy season, where the
relatively more concentrated rainfall may directly form surface runoff and rapidly recharge the
river.

In the Eastern Kunlun Mountains water system, the spatial trend of river water H-O isotopes

depletion from west to east (Figure 4) elucidates the variation of precipitation isotopes in relation
to the water vapor transport process, which can be attributed to the waning of westerly winds.
Along the water vapor transport path, heavy isotopes are preferentially separated in precipitation
formation, leading to an augmentation in the continental characteristics of water vapor carried by
the air mass, while the precipitation formed by the remaining water vapor undergoes a gradual
depletion of isotopes (Yang and Wang, 2020). Meanwhile, the two watersheds in the Qilian
Mountains possess contrasting spatial variation characteristics relative to the Eastern Kunlun
Mountains water system. Based on the comparison and analysis of the meteorological elements of
Delingha and Da Qaidam (refer to Figure 2 for specific location) from 2010 to 2020, the average
annual precipitation of Delingha (276.36 mm) was 2.41 times higher than that of Da Qaidam
(114.79 mm), and the average annual temperature of Delingha (5.23 °C) was higher than that of
Da Qaidam (3.65 °C) by 1.58°C. Since 1961, precipitation in the Bayin River has risen by as much
as 25.09 mm/10 a, which was more than six times greater than that of the Yuka River. Owing to
the abundance and marked magnitude of increase of precipitation, the seasonal $\delta^{18}$O variation in
the Bayin River was approximately 1.79 times that of the Yuka River. Under similar conditions
where ice/snow meltwater recharge was present, the mean $\delta^{18}$O and $\delta$D values of the Bayin River
were positively skewed by 1.52‰ and 7.26‰ relative to that of the Yuka River, respectively,
which can be attributed to the greater contribution of precipitation with heavy isotopic enrichment





characteristics to the river water. Therefore, the spatial and seasonal variations of the river water
H-O isotopes in the Qilian Mountains water system can be attributed to the variations in
hydrothermal conditions and the varying extents of warming and humidification in the watershed.
To further explain the cause of seasonal and spatial variations of surface water $\delta^{18}O$ and $\delta D$
values, ERA5 reanalysis data in the rainy season (June to September) were used to calculate the
water vapor flux field in the Qaidam Basin and its surrounding areas and track the main paths of
the water vapor transport of precipitation (Hersbach et al., 2019). The results (Figure 6)
demonstrated that the water vapor path in and surrounding the basin is predominantly affected by
the mid-latitude westerly air masses (Yang and Wang, 2020) and the water vapor flux in the eastern
region of the basin is notably greater than that in the western region. This explains the spatial
change patterns of river water H-O isotopes (Figure 4) and hydrothermal conditions to a large
degree (Figure S1). The above findings are also supported by atmospheric and isotopic tracing
evidences. For example, the Tanggula Mountains (33°–35° N) form the physical and chemical
boundary of the Tibetan Plateau, and the northern region is fundamentally under the control of the
westerly wind, which hinders the South Asian monsoon from exerting a direct influence on the
Qaidam Basin (Yao et al., 2013; Kang et al. 2019; Wang et al., 2019). Furthermore, water vapor
source information can be reflected in d-excess, where the recycled moisture that evaporated under
conditions of low humidity and water carried by the westerly wind is considered to possess higher
d-excess values. The mean d-excess of basin river water samples during the wet season (11.45‰,
Table S1) was greater than 10‰, which reflects the characteristics of an alpine arid continental
climate and a water vapor source devoid of monsoon influences. In contrast, in the hinterland of
the Tibetan Plateau, south of the Tanggula Mountains, which is subject to considerable influences
from the South Asian monsoon circulation, the d-excess values of summer precipitation and river
water were in the range of 5‰–9‰, with mean value of 7‰ (Tian et al., 2001). The substantial
differences in the d-excess values between the two regions also support the above inference about
the water vapor sources of the Qaidam Basin.





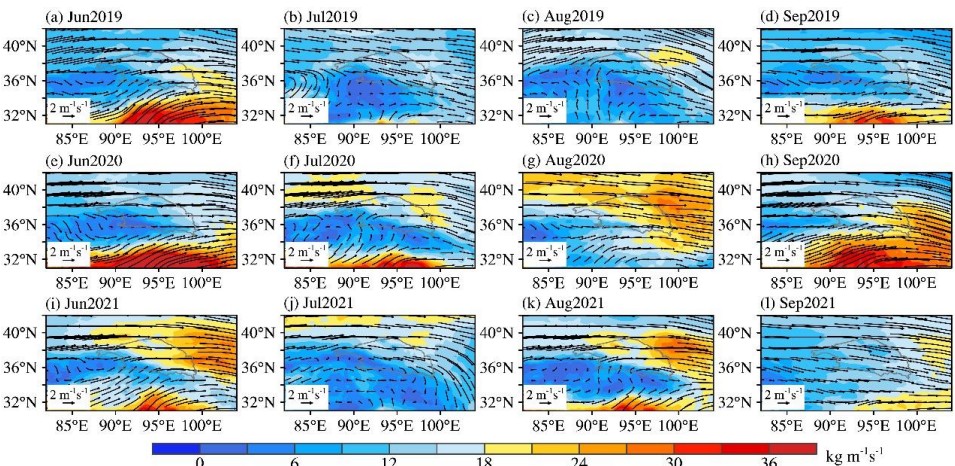


**Figure 6.** Tropospheric water vapor flux from June to September 2019 to 2021 (below 500 hPa, unit: kg m$^{-1}$ s$^{-1}$)

Given the spatiotemporal differences of surface water $\delta^{18}$O-$\delta$D (Figure 4), samples of
different water bodies in each watershed were incorporated into the $\delta^{18}$O-$\delta$D plot (Figure 7). The
presence of considerable differences in isotope distribution characteristics suggests that seasonal
changes in the surface water H-O isotopes in each water type may be due to differences in the
proportions of the contribution of precipitation, ice/snow meltwater, and groundwater during the
wet and dry seasons. Hence, Equation 1 (Table 1) was employed to estimate the contribution of
each potential recharge endmember to the river water. The findings indicated that in the dry season,
the base flow in each watershed is maintained by the groundwater discharge in mountainous areas,
where the contribution of groundwater to base flow may reach up to 97%. During the wet season,
the river water in each watershed is recharged by different proportions of precipitation, ice/snow
meltwater and groundwater. For example, in the area with the most abundant annual rainfall, the
contribution of precipitation to the Bayin River may reach 84% during the wet season. The
westerly water vapor forms more precipitation as a result of obstruction from landforms in the
eastern region of the basin, in contrast, as the source area of the Bayin River is in close proximity
to the eastern region of Qilian Mountains, although the summer monsoon is likened to 'an arrow
at the end of its flight' (a spent force), the latter continues to contribute more than 22% of the
oceanic water vapor to the nearby areas (Wu et al., 2022). The topographic obstruction and strong
convection form abundant precipitation, rendering the proportion of precipitation in the surface
runoff in the eastern basin region appreciably higher than that in other areas. Thus, differences in





the proportions of contribution ratio of each recharge end member during wet and dry seasons are
the main factors responsible for the seasonal variations in surface water isotopes in each watershed.

In summary, the seasonal variations and spatial patterns of surface water stable H-O isotopes

are a consequence of the combined effects of the extent of warmth and humidity in the region,
intensity of the mid-latitude westerly wind water vapor transport, and local hydrometeorological
conditions.



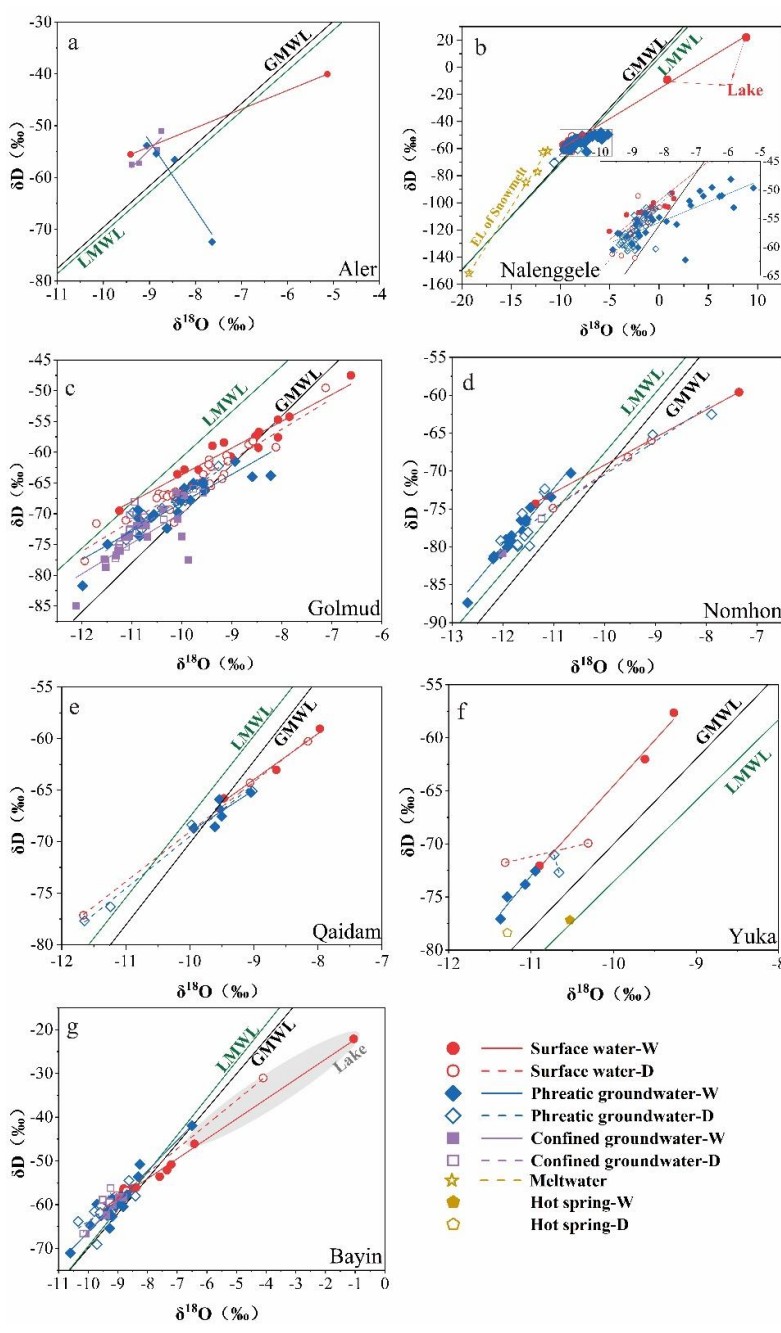

**Figure 7.** δ¹⁸O-δD plots during dry-wet seasons and in different water bodies in each watershed of the Qaidam Basin (W and D represent wet and dry seasons, respectively)





**Table 1.** Contribution ratios of endmembers to river water during the wet and dry seasons based on δ18O and d-excess (Unit: %; W and D represent wet season and dry season, respectively)

| | Endmember | Groundwater | Meltwater | Tributary | Precipitation |
|---|---|---|---|---|---|
| Nalengele-W | Mean | 0.41 | | 0.47 | 0.12 |
| | Max | 0.60 | | 0.74 | 0.13 |
| | Min | 0.18 | | 0.27 | 0.08 |
| | SD | 0.12 | | 0.13 | 0.02 |
| Nalengele-D | Mean | 0.90 | 0.10 | | |
| | Max | 0.97 | 0.27 | | |
| | Min | 0.73 | 0.03 | | |
| | SD | 0.07 | 0.07 | | |
| Golmud-W | Mean | 0.31 | 0.34 | 0.25 | 0.10 |
| | Max | 0.36 | 0.39 | 0.32 | 0.12 |
| | Min | 0.28 | 0.29 | 0.20 | 0.08 |
| | SD | 0.03 | 0.04 | 0.05 | 0.01 |
| Golmud-D | Mean | 0.32 | 0.25 | 0.42 | |
| | Max | 0.46 | 0.45 | 0.70 | |
| | Min | 0.19 | 0.11 | 0.21 | |
| | SD | 0.09 | 0.10 | 0.17 | |
| Yuka-W | Mean | 0.62 | 0.23 | | 0.15 |
| | Max | 0.76 | 0.29 | | 0.18 |
| | Min | 0.55 | 0.15 | | 0.10 |
| | SD | 0.10 | 0.06 | | 0.04 |
| Bayin-W | Mean | 0.26 | 0.04 | 0.25 | 0.45 |
| | Max | 0.35 | 0.05 | 0.43 | 0.84 |
| | Min | 0.08 | 0.02 | 0.06 | 0.23 |
| | SD | 0.08 | 0.01 | 0.11 | 0.19 |

5.2 Sources and spatial patterns of groundwater recharge

The seasonal variations in groundwater aquifer H-O isotopes in each watershed suggests that differences exist in their recharge sources, forms, and rates. The δ¹⁸O-δD relationship of different seasons and different types of water samples can be used to elucidate the groundwater source composition and recharge pattern. The Qaidam Basin groundwater system can be classified into three types of recharges according to the seasonal changes in groundwater δ¹⁸O-δD in each watershed (Figure 5) as well as the δ¹⁸ O-δD relationship of different water bodies within the watershed (Figure 7).





### 5.2.1 Heavy precipitation in the wet season-dominated recharge

In the Nalenggele River, which is situated in the southwestern basin, and the Qaidam and Bayin Rivers in the eastern basin, groundwater $\delta^{18}O$ and $\delta D$ values were markedly positively skewed during the wet season. The groundwater isotope data distribution in the majority of the wet seasons was closer to the LMWL and GMWL compared to that during the dry season. Moreover, the isotopic characteristics were closer to the river water and summer precipitation in the same period (Table S1; Zhu et al., 2015), with different trends in evaporation (Figures 7b, 7e and 7g). These results indicate the contribution of precipitation to groundwater during the wet seasons. The relatively marked seasonal variations of H-O isotopes also demonstrate that the aquifers in the eastern and southwestern Qaidam Basin have a relatively rapid groundwater cycle and present seasonal recharge. In the eastern basin, there is an abundance and notable rise in precipitation (Figure S1), which has directly led to a rise of 5 m in water level and surface area expansion of 1.59 times in a lake at the source of the Nalenggele River in the southwestern basin from 1995 to 2015 (Chen et al., 2019). This further indicates the abundant precipitation in the headwater may be a potential source for the rapid seasonal recharge of groundwater recharge associated with the rapid climate warming and humidification. Additionally, the tectonic conditions of the recharge area are factors that are in favor of driving seasonal groundwater recharge. These three watersheds happened to be situated in the collision zone (Figure 2), where neotectonic movement is strong, and there are substantial developments of deep fractures and faults in the recharge area. It can be inferred from the aforementioned that favorable hydrological and tectonic conditions promote the formation of direct and rapid recharge of groundwater through bedrock fissures under the large hydraulic head (>1000 m) from precipitation and meltwater at higher altitudes, resulting in substantial seasonal changes in the groundwater H-O isotopes in these regions (Tan et al., 2021).

### 5.2.2 Glacial-snow melt water-dominated recharge

In the Nomhon and Yuka Rivers situated in the central basin region, groundwater H-O isotopes were more depleted during the wet season compared to the dry season (Table S1; Figure 5). Figures 7d and 7f also show that the majority of the $\delta^{18}O$–$\delta D$ data for the groundwater samples in these two watersheds were observed in the lower left of the LMWL and GMWL. These values were far away from the LMWL and GMWL, and were more negatively skewed relative to river





water, with characteristics being closer to the snow meltwater observed in the high-altitude Eastern
Kunlun Mountain (Figure 3; Yang et al., 2016). This demonstrates that the groundwater recharged
by ice/snow meltwater was more depleted in heavy isotopes during the wet–dry seasons and the
contribution of precipitation to the aquifer was relatively small. Similarly, on the eastern margin
of the Tibetan Plateau, the phenomenon where the non-monsoon meltwater controls the monsoon
groundwater system hydrological process has been observed (Kong et al., 2019). The isotope
signals indicated that ice/snow meltwater depleted in the heavy isotopes in the source area was
released as a result of the rising temperature in summer, and following the mixing of groundwater
with the seasonal meltwater recharge, the groundwater was further depleted in heavy isotopes.
Furthermore, owing to the low precipitation in these two watersheds (61.39 and 121.78 mm, Figure
S1), the precipitation in 2020 was even lower. Under extremely arid climate, the direct recharge
to the aquifer from the limited precipitation was negligible. GRACE data also showed that the
melting of solid water in the source area due to climate warming was a key factor driving the
increase in the groundwater storage of the Qaidam Basin (Xiang et al., 2016). This further supports
the inference that groundwater isotopic depletion during the wet season stems from the seasonal
melting recharge of the cryosphere.
### 5.2.3 Fossil water-dominated recharge
In the Golmud River, the mean $\delta^{18}O$ value during the wet season was 0.33‰ higher than that
during the dry season and seasonal changes were not apparent, indicating that the proportion of
seasonal groundwater recharge was small and the renewal rate was slow. The H-O isotope data of
the groundwater were mainly located between the LMWL and GMWL (Figure 7c), indicating that
the main recharge source was the atmospheric precipitation of different seasons. In addition, the
groundwater $\delta^{18}O$ and $\delta D$ in the alluvial fan belt exhibited a gradually negatively skewed trend
along the flow path (Figures 8a and 8b). A prominent feature of this watershed is the sizeable
storage of confined groundwater, which is continuously emanating at the front edge of the alluvial
fan. Confined groundwater possesses $\delta^{18}O$ and $\delta D$ values that are more negative than phreatic
groundwater, and the mean $\delta^{18}O$ values during the wet–dry seasons are consistent, without any
seasonal changes. The substantial differences in isotopic characteristics between phreatic and
confined groundwaters (Table S1) suggest that there are potential differences in their recharge
sources. It is speculated that phreatic groundwater is predominantly recharged by ice/snow





meltwater while confined groundwater is slowly and stably recharged and may be sustained by
precipitation with low δ¹⁸O and δD values or fossil water formed under the relatively cold climatic
conditions (Xiao et al., 2018).





**Figure 8.** Spatial distribution of groundwater $\delta^{18}$O (a) and $\delta$D (b), and surface water–groundwater $^3$H (c) concentrations during the wet season (Circle and asterisk represent groundwater and surface water, respectively)

5.3 Water cycle mechanism

Being a direct constituent of water molecules, radioactive $^3$H with a half-life of 12.32 years can be used to estimate the migration time of younger water. Particularly, $^3$H can effectively trace groundwater age and renewal rate in water bodies consisting of a mix of younger water and fossil water (Xiao et al., 2018; Chatterjee et al., 2019; Shi et al., 2021). In accordance with the considerable differences in $\delta^{18}$O-$\delta$D of the different water bodies in each watershed (Figures 7, 8a and 8b), the scale and extent of the groundwater recharge in the Qaidam Basin were further determined using the $^3$H concentration. The $^3$H concentration spatial distribution pattern indicated that there were considerable differences in groundwater recharge rates at both intra- and inter-watershed scales (Figure 8c). Thus, the groundwater system was dominated by both regional and local recharge.

At the watershed scale, the $^3$H concentration of the phreatic groundwater in the alluvial fan zone near the river channel and mountain pass was considerably higher (Table S1) and close to that of the river water. These results indicate that close hydraulic interactions occurred between the surface water and groundwater and the aquifer also receives river water via vertical infiltration and lateral runoff recharge. Hence, this portion of groundwater is dominated by seasonal modern water recharge, which is younger and has a relatively more rapid renewal rate. The phreatic and confined groundwater $^3$H concentrations at the edge of the alluvial fan were largely below 3 TU or lower than the detection limit, which was inconsistent with that near the river channel. These findings suggest that these aquifers are predominantly recharged by lateral runoff, which consists largely of submodern water (>60 years) or fossil water, and the mixing proportion of modern precipitation and seasonal meltwater is relatively low, with a slow renewal rate. This situation is particularly apparent in Golmud and Nomhon Rivers (Liu et al., 2014; Cui et al., 2015; Xiao et al., 2017, 2018), which also further reflects the importance of fossil water content in maintaining the aquifer in extremely arid regions.

At the basin scale, radioactive $^3$H concentration characteristics are consistent with the water cycle information indicated by the seasonal changes in stable H-O isotopes. In the phreatic groundwater systems situated in the eastern and southwestern basin regions, $^2$H and $^{18}$O are





relatively enriched during the wet season, $^3$H has a relatively higher average concentration,
seasonal groundwater recharge is more noticeable, and groundwater is overall younger (<60 years).
Based on river seepage, modern meltwater and precipitation may also infiltrate through favorable
structural water passage channels, such as fault zones developed on a large scale in the recharge
area, resulting in rapid recharge to the aquifer (Figure 9b; Tan et al., 2021). The phreatic
groundwater systems in the western Qilian Mountains and central Eastern Kunlun Mountains were
relatively depleted in $^2$H and $^{18}$O during the wet season, the $^3$H concentrations were
correspondingly low, and these systems mainly received recharge from seasonal ice/snow
meltwater. However, due to the relatively stable recharge of meltwater by comparison, the
groundwater renewal rate was relatively slow (Figure 9c).
The depletion of heavy H-O isotope was greatest in confined groundwater, and the $^3$H
concentration of the majority of the samples was very low (<3 TU) or fall below the detection limit,
which indicates that the confined groundwater recharge rate is very slow. Furthermore, the
confined groundwater was largely over 100 years old and consisted predominantly of submodern
groundwater or fossil water (Xiao et al., 2018). In the Golmud River, the effect of H-O isotope
seasonal changes in most of the confined groundwater was relatively small or showed almost no
detectable change. However, the confined groundwater in the overflow zone continued to
spontaneously emanate after nearly half a century of mining and the water pressure did not
decrease, which suggests that modern precipitation or ice/snow meltwater recharges deep confined
groundwater. Some confined groundwaters possess recognizable isotopic seasonal effects, and the
existence of a certain proportion of continuous recharge, even on a seasonal scale, cannot be ruled
out. In addition, large karst springs have developed in the mountainous areas of the Golmud River.
Moreover, a large karst spring (KLSQ-1) was observed near the mountain pass, with a flow rate
as high as 224.7 L/s. In mountainous areas, well-formed karst cavities and fissures provide
conduits to enable the direct infiltration of precipitation or meltwater. Following deep circulation,
precipitation and meltwater give rise to regional subsurface runoff, which recharges confined
groundwater in the overflow zone over a long distance, thus causing it to flow continuously under
the effect of a large hydraulic head (approximately 1,000 m) (Figure 9d). Moreover, in the Golmud
and Bayin Rivers, the H-O isotopic signals of some confined groundwaters at the front edge of the
alluvial fan were essentially consistent with that of the nearby phreatic groundwater and the $^3$H





concentration was close to 10 TU. These findings also suggest that the confined groundwater may
pass through the aquitard or leakage recharge occurs in the local skylights.

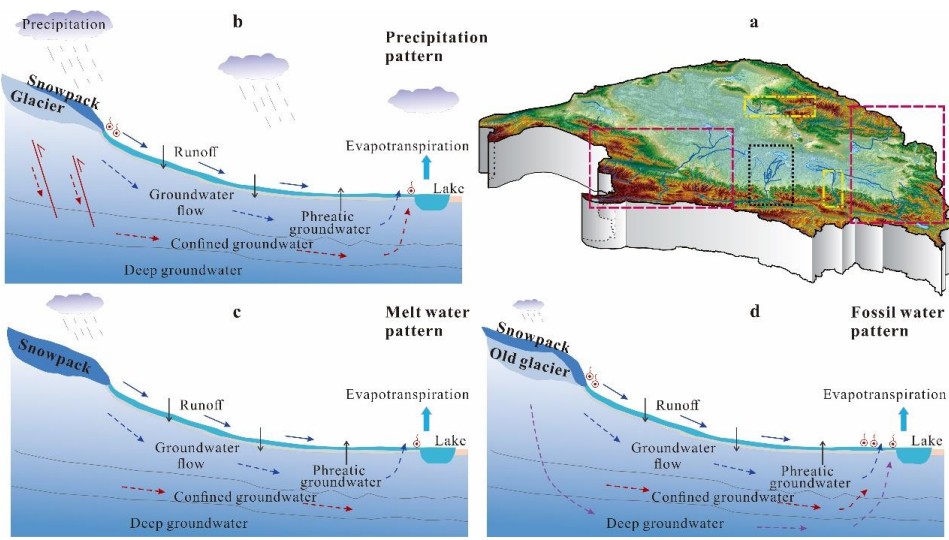


**Figure 9.** Schematic diagram of the Qaidam Basin water cycle model (b represents the blue dashed box; c
represents the yellow dashed box; and d represents the black dashed box)

5.4 Isotope hydrology responses and water cycle trends under climate change

Since the 1980s, the Qaidam Basin has experienced rapid warming at a rate more than twice

the global average (Wang et al., 2014; Kuang and Jiao, 2016; Yao et al., 2022). Since 1960, average
annual temperature and precipitation variations and increasing rates over every 10 and 30 years at
eight meteorological stations in the basin (Figure 10a) and continuous air temperature and
precipitation change trends at two representative meteorological stations in the north and south
(Delingha and Golmud) (Figure 10b) have shown that the present warming and humidification
trends in the northeastern Tibetan Plateau are continuously strengthening. Previously, the general
assumptions were that the isotopic composition of the surface water and groundwater systems did
not vary with time, at least on interannual scales, and was relatively stable (Boutt et al., 2019).
However, over the past 40 years, the isotopic variations of water bodies have demonstrated that
varying degrees of interannual differences in surface water and groundwater isotopes exist and
interannual variations in the average $\delta^{18}O$ value are greater than 3‰ (Figure 11). Therefore,
isotopic changes reflect different extents of sensitivity to climate change, regardless of the seasonal
or multi-year scale. The spatiotemporal variability of isotopic signals can be ascribed to differences

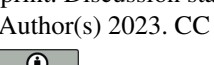



in the extent of warming and humidification in each watershed. Wang et al. (2014) highlighted
that while the Qaidam Basin has experienced rapid warming in the past 50 years, the extent of
warming and humidification in different regions is noticeably not in sync, where the rate of
temperature rise ranged from 0.31 to 0.89°C/10a and the rate of rainfall increase ranged from 1.77
to 25.09 mm/10a (Figure S1). The correlation between $\delta^{18}O$ of watershed surface–groundwater
and temperature and precipitation showed (Figure 12) that the multi-year scale $\delta^{18}O$ variations in
the basin surface water and groundwater had a more significant positive correlation with
precipitation in the same period than temperature (Figures 12b and 12d). Of note, surface water
was more sensitive to precipitation (Figures 12a and 12b) while groundwater was more sensitive
to temperature (Figures 12a and 12c). This phenomenon suggests that the rise in rainfall may affect
the water cycle by promoting slope runoff and groundwater infiltration in mountainous regions
and indicates that warming will lead to the ablation of solid water at higher altitudes to accelerate
the groundwater recharge of aquifers through bedrock fissures. The GRACE and remote sensing
monitoring findings also demonstrated that the increase in terrestrial water storage in the Qaidam
Basin is closely linked to the rise in rainfall and glacier meltwater recharge (Song et al., 2014; Jiao
et al., 2015; Xiang et al., 2016; Wei et al., 2021; Zou et al., 2022), which fully supports the isotope-
based conjecture. Furthermore, recent research demonstrates that the accelerated conversion of ice
and snow on the Tibetan Plateau into liquid water has led to an imbalance in the "Asia Water
Tower", with the Qaidam Basin one of the major regions experiencing an increase in liquid water
(Yao et al., 2022). The consistency of data on H-O isotopes, remote sensing and hydrometeorology
shows that the Qaidam Basin is a region in the Tibetan Plateau with the most rapid and substantial
warming. Global warming affects the basin by causing a redistribution of precipitation and melting
ice and snow in high-altitude areas, resulting in a rise in groundwater storage and an expansion of
the area of lakes, among other effects. Additionally, the corresponding rise in precipitation in the
mountainous areas is able to supplement the rapidly melting ice and snow to a certain degree. This
trend of water storage increase in the Qaidam Basin is likely to continue in the 21st century. The
highly coupled findings of different observation methods further emphasize the sensitivity and
potential of water stable isotopes in tracing water cycles and climate change.

Due to the effects of climate change and the intensification of cryosphere retreat, runoff has

changed considerably on the Tibetan Plateau, which drastically impacted the spatiotemporal
distribution of water resources (Wang et al., 2021). Based on our observation results, it can be





speculated that with continued rapid warming and humidification, the water resources of the watershed with substantial seasonal recharge may manifest as follows: The amount of surface water and groundwater resources will considerably increase in the short term (in recent decades) because of the shift of snow line and rapid melting of ice and snow coupled with the increase in precipitation. For example, in the Bayin and Qaidam Rivers in the eastern basin, as a result of the abundant and marked increase in precipitation and strong water resource renewal capability, the water reserves may sustain an increasing trend in the long term under the influence of continuous climate warming. This phenomenon has been verified in many regions of the Tibetan Plateau and some alpine watersheds in high-latitude Switzerland (Xiang et al., 2016; Malard et al., 2016; Shi et al., 2021). Moreover, the cyclical nature of climate change also suggests that cryosphere retreat on a large scale may not be sustainable for watershed surface–groundwater recharge. It was reported that the glaciers in the southwestern basin are continuously losing mass (−0.2 to −0.5 m/a), and this trend has substantially increased from 2018 to 2020, particularly in the headwaters of Nalenggele River, where the glacier elevation has reduced by 5.42 m since 2000 (Shen et al., 2022). However, as a result of the low precipitation in the southwestern basin, achieving equilibrium in recharge remains a challenge given the rapid melting of ice and snow caused by climate warming, even if precipitation continues to increase in the future. This means that in the future climate change scenario, water resources in the southwestern basin watershed (such as Nalenggele River) may continue to rise for a certain period before showing a large-scale decrease. This trend of initial increase followed by decrease is common in the Tibetan Plateau or regions with relatively little precipitation in alpine watersheds around the world. Furthermore, in the watersheds of the central basin (Nomhon, Golmud, and Yuka Rivers), while the surface–groundwater recharge is relatively stable, the long-term large-scale exploitation of groundwater in these three areas during the industrial and agricultural development processes has decreased the precipitation in the source area and led to a reliance on ice and snow melting. Moreover, a decline in groundwater level fluctuations in the future is inevitable. Data monitoring of the five shallow groundwater boreholes in the alluvial fan belt of the Golmud River showed that the groundwater level has fluctuated and reduced by an average of −1.18 m/a since 2011 (Figure S2). Whether the increase in water resource renewal capacity and water storage in the Qaidam Basin can remain stable is a scientific issue that is worthy of consideration in the future.



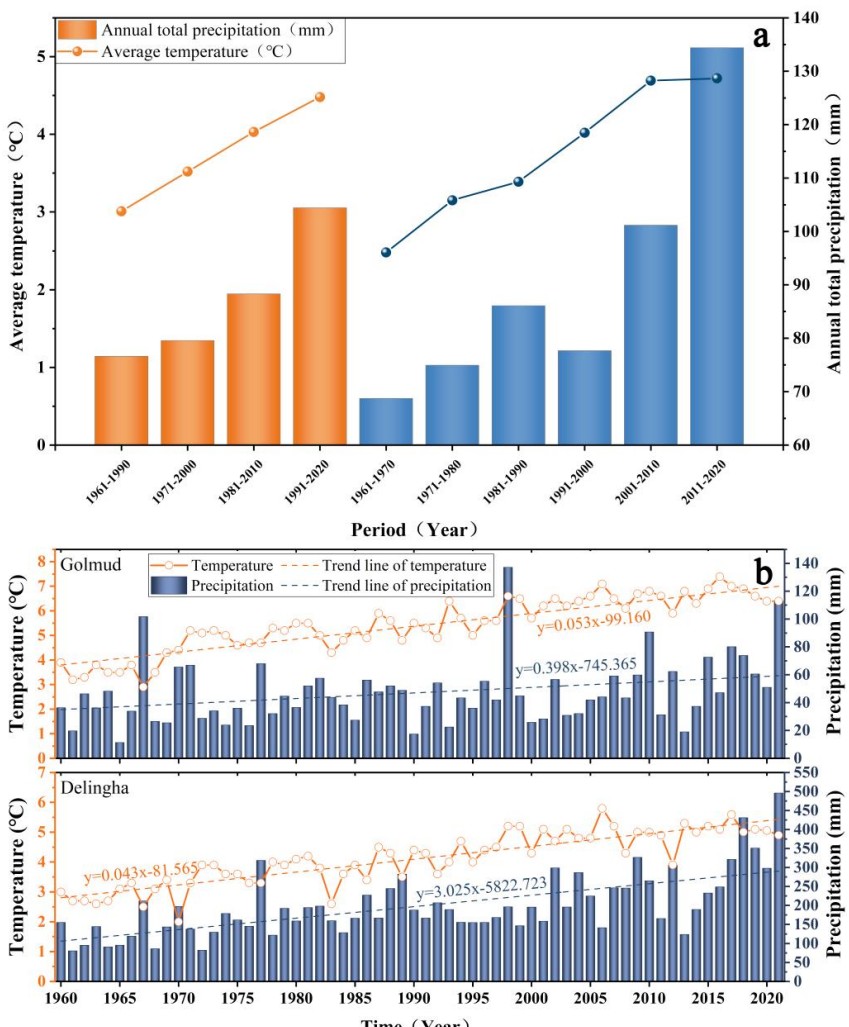

**Figure 10.** Average temperature and precipitation in the Qaidam Basin every 30 years and 10 years from 1960
to 2020 (a); interannual changes in temperature and precipitation at the Golmud and Delingha meteorological
stations (b)


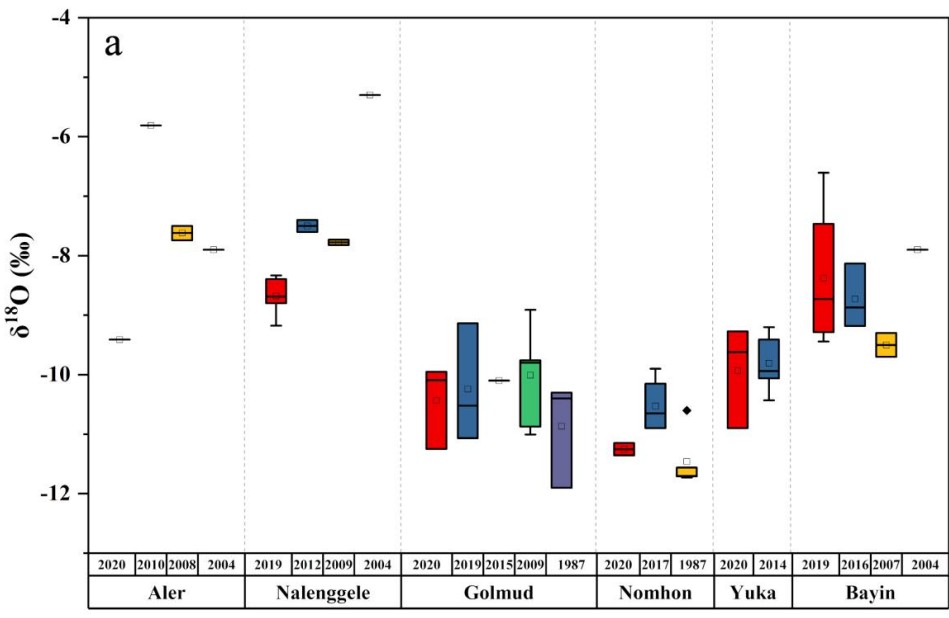

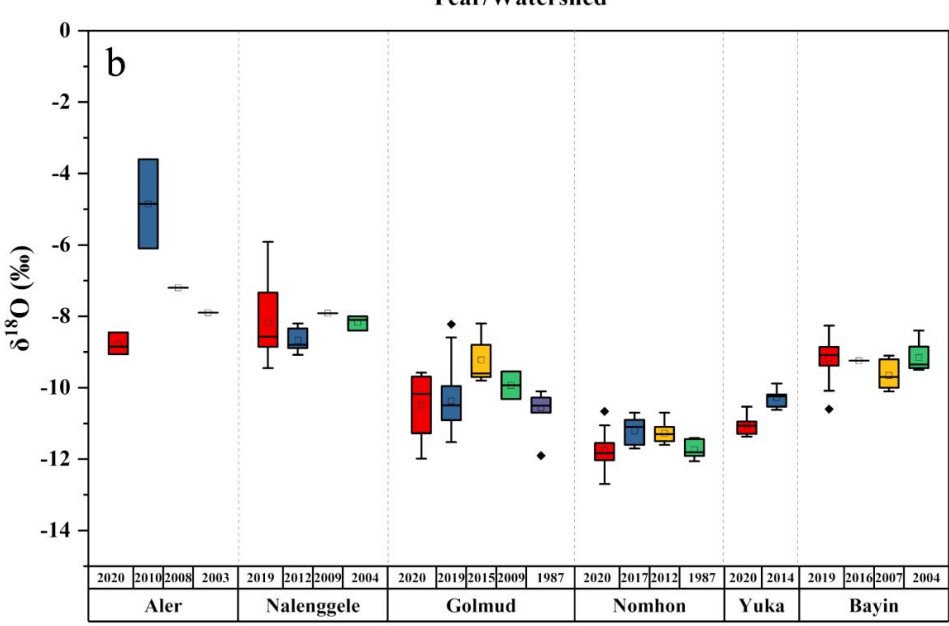

**Figure 11.** Interannual variations in the river water and groundwater δ¹⁸O in the Qaidam Basin



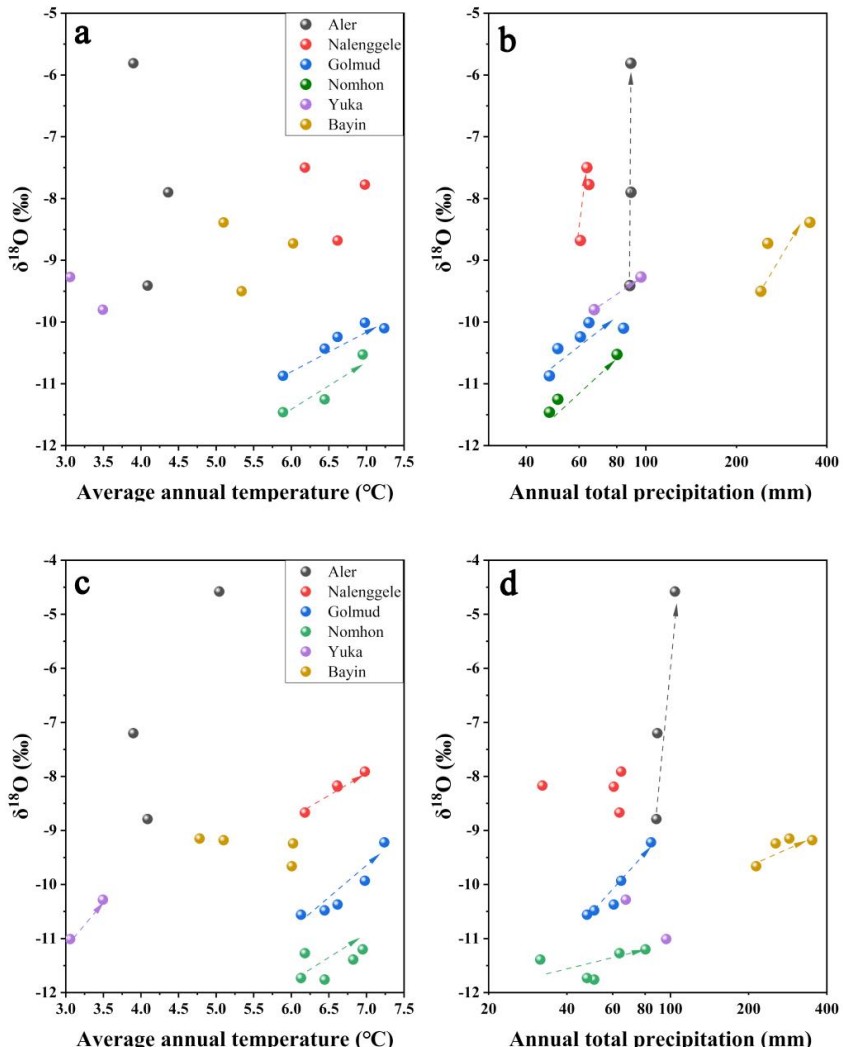

**Figure 12.** Surface water $\delta^{18}$O and temperature (a) and precipitation (b); Groundwater $\delta^{18}$O and temperature (c) and precipitation (d) in the Qaidam Basin

## 6. Conclusion

(1) The contribution of precipitation and ice/snow meltwater is the main factor that drives the accelerated water cycle in the Qaidam Basin. The spatiotemporal variations of surface water and groundwater $\delta^{18}$O and $\delta$D reflect their dynamic responses to water sources, climate warming, and neotectonic movements, especially precipitation, at interannual and seasonal scales. Surface water H-O isotopes are enriched during the wet season and relatively depleted during the dry season with



a remarkable evaporation effect. The mean values of surface water $\delta^{18}O$ and $\delta D$ in the Eastern Kunlun Mountains water system are gradually negatively skewed from west to east, and the reverse holds true for the Qilian Mountains water system. The seasonal differences of isotopes are determined by the precipitation level and its increase in the watershed, and the spatial change patterns reflect the influence of water vapor transport intensity of the westerly path and local climatic conditions. The base flow is maintained by groundwater recharge during the dry season, and varying proportions of groundwater (26%–62%), ice/snow meltwater (23%–47%) and precipitation (10%–45%) are received during the wet season. The contribution of precipitation to surface rivers in the Qilian Mountains is greater than that of the Eastern Kunlun Mountains.

(2) The phreatic groundwater system located in the collision and convergent zone of different mountain ranges is characterized by enriched H-O isotopes, high concentrations of radioactive $^3H$, and marked seasonal recharge during the wet season. Modern meltwater and precipitation are able to infiltrate through favorable structural water channel passages, such as large-scale active fault zones, giving rise to rapid groundwater recharge. In contrast, the phreatic groundwater systems in the western region of the Qilian Mountains and the central region of Eastern Kunlun Mountains have depleted H-O isotopes and low $^3H$ concentrations during the wet season, and they are mainly slowly recharged by seasonal ice/snow meltwater, which consists of modern water and submodern water (>60 years) maintained together. The confined groundwater is considerably depleted in H-O isotopes, and a majority has no apparent seasonal changes. The $^3H$ concentration is very low or below the detection limit, and the recharge is relatively slow with fossil water dominated.

(3) Climate warming has exerted a substantial impact on the hydrological processes throughout the whole basin, thereby driving water cycle acceleration and increases in water storage fluctuations in the eastern and southwestern basin regions via the increase in precipitation and melting of glaciers and snow. However, the cyclical nature of climate change suggests that this trend is unsustainable. As precipitation increases and solid water ablation in mountainous regions becomes severely out of balance, the southwestern basin may face a rapid decline in total water resources in the future.



**Author Contribution**

Conceptualization: Yu Zhang, Hongbing Tan; Funding acquisition: Xiying Zhang; Investigation: Peixin Cong; Resources: Wenbo Rao; Visualization: Dongping Shi; Writing– original draft: Yu Zhang; Writing–review & editing: Hongbing Tan.

**Acknowledgments**

This study was financially supported by the National Natural Science Foundation of China (U22A20573) and the Postgraduate Research & Practice Innovation Program of Jiangsu Province (KYCX22_0666). We would like to express our gratitude for all members' help both in the field observation and geochemical analysis in the laboratory.

**Declaration of interests**

The authors declare that they have no known competing financial interests or personal relationships that could have appeared to influence the work reported in this paper.

**Data Availability Statement**

The complete list of isotopes and their values is available in Table S1 in Supporting Information. The meteorological data can be obtained on China Meteorological Data Network (http://data.cma.cn). The monthly mean ERA5 reanalysis data ($0.25° \times 0.25°$) can be obtained from European Centre for Medium-Range Weather Forecasts (ECMWF, https://www.ecmwf.int/).

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

dominates the increase of total groundwater storage in the Tibetan Plateau. Geophysical
Research Letters, e2022GL100092.
