# Peer review of "Isotopic variations in surface waters and groundwaters of an extremely arid"

_Hydrology and Earth System Sciences, 2023_

## Referee Comment (RC2)

General comments

This paper aims to reveal the response of hydrological processes to climate change in an extremely arid basin located on the north-eastern Tibetan Plateau, where it is really an ideal area to evaluate hydrological response to climate changes. With the support of large datasets including stable isotopes and water chemical compositions for a whole basin and its around mountains from different spatial-temporal scales, the manuscript has clarified the entire water cycle processes as well as the isotope hydrology responses to rapid climate changes in an extremely arid basin in the Tibetan Plateau. In addition, the manuscript has also concluded a perspective on the variation trends of water resources under the condition of multiple water sources recharge and climate warming. As my assessment, the topic about the impact of climate change on water resources talked in this manuscript is a general global focused issue. The research objectives, methodology, content and scientific hypotheses of this manuscript are well coincided within the scope of HESS. Thus, I suggest to accept this manuscript, after somewhat improvements. My main concerns are as follows:

1. The title needs to be reconsidered. A title with a clear direction would help to highlight the topic. I believe that the authors may want to emphasize the response of surface water and groundwater isotope variations to climate change. In addition, some of the sub-headings in the text should be also reconsidered so that they can outline the content of the text.

2. It seems to me that the abstract and introduction need to be furtherly condensed. Although the article is now look -like systematic, a more focused topic is the key to attract the reader's attention.

3. As for the text structure, it seems to be a bit confused and may be necessary adjusted. For example, the Section 4.1 should to be better to move after Sections 4.2 and 4.3. It seems to be too long for section 5.1, some subheadings may be added in order to help to fill the gap for readers. Section 5.3 is mainly focused on the groundwater circulation mechanism, which in my view, it should be moved to section 5.2. In addition to the overall structure, the content between paragraphs also needs to be sorted out. Example, the second paragraph of section 5.4 is too long and includes too many ideas. It should be rewritten to be more in point or logical.

4. Some Figures need to be rehearsed, e.g., regarding Figure 2, a hydrogeological map may be more appropriate for the topic of the paper rather than a simplified tectonic map. In addition, detail information needs to be added in some figure captions.

Some minor shortcomings please consider to improve:
Lines 13-14: Delete "The surface water heavy isotopes enrich during the wet season and deplete during the dry season". The abstract should highlight key findings.

Line 19: hydrothermal conditions are not an appropriate terminology, I guess the author wants to express temperature and precipitation regimes.

Line 21: Delete "ice/snow".

Line 28 Keywords should be ordered according to major and second categories and then followed specific research directions

Line 37:Delete "region of the".

Line 57: "These issues require an in-depth investigation" is a repetition of the "several questions remain to be resolved" in Line 55.

Lines 91-95: No more than 3 goals of the study were considered reasonable. The authors were advised to streamline the study objectives. The order of the research objectives should also be expressed more logical.

Line 118: "Hydrogeology and structure" can be changed to Basic hydrogeological setting

Line 137: Altun is not Alun

Line 162: phreatic groundwater is not phrenic groundwater

Line 206: Section 4.1 seems to duplicate 4.2 and 4.3 in terms of structure, although the contents are different. I think the contents of 4.1 can be moved after 4.2 and 4.3 after simplification.

Line 245: Specific terminology in the whole text should be used, e.g."positively skewed" is not a terminology and it easily causes ambiguity. For isotopic expression, please use "depleted or enriched" to decipher their variations.

Line 254: "seasonal and spatial variations" should be replaced by "spatial and seasonal variations".

Line 263-268: The description of spatial variations should be placed before the description of temporal variations in order to match the title.

Line 292: In chapter 5.1, some subheadings need to be added. Also, the order of the paragraphs should be logically adjusted. The paragraph on moisture tracing should be placed previously, and the content of Lines 309-330 should be swapped with Lines 356-376.

Line 377: please revise it as Line 254.

Line 378: According to my understanding, regional warming and humidification trends is better than the extent of warmth and humidity in the region

Line 386: The headings and subheadings of chapter 5.2 also need to be reconsidered to summarize the topic of the paragraph. Additionally, in this chapter, the main idea of the article should be strengthened but some discussions that are not closely related to the scientific issues should be shorten or deleted.

Line 394: 5.2.1 In the extremely arid region, recharge from precipitation occurs mainly during the wet season, which is a general idea and need not to be emphasized.

Line 445: Can the authors please confirm that it is DIFFERENT SEASONS that is correct? I think the author meant different phases.

Lines 502-504: These sentences can be deleted for repetition in the result section.

Lines 651-562: This is a repetition of the previous sentence.

Line 599: Figure 10b is redundant, although it presents a lot of data, as Figure 10a is sufficient to show the striking climate change trends in the study area. It could also be removed from the text, as the obvious redundancy of information is not advocated.

Line 604: The authors should report the source of the data in Figure 11. Because some of the data are obviously collected from the literature. I would suggest putting figure 11 in the annex as the highlights of this section are already shown in Figure 12.

---

## Author Response (AR1)

Dear Prof. Markus Hrachowitz,

Thank you for your time and attention in handling our manuscript, as well as for providing us an opportunity to revise and improve its quality. We are pleased to submit a revised version, now entitled, "***Isotopic variations in surface waters and groundwaters of an extremely arid basin and their responses to climate change***". The manuscript has been substantially revised according to three reviewers' comments and suggestions. In addition, Prof. Beckie, R. D., University of British Columbia, an accomplished scientist in the field of Hydrogeology, has also thoughtfully checked this revision for language as well as scientific issues. We believe that the revised version is now very different from the initial, both with respect to organization, scientific and data presentation and the English language or grammas. We would like to sincerely thank you, three reviewers and Prof. Beckie, R. D. for these valuable comments. In particular, we have significantly restructured the manuscript framework based on comments, especially the discussion section, to highlight our focus. Included are responses to comments from the three reviewers, a manuscript with tracked changes and a manuscript with no marks. The following are point-by-point responses to the reviewers' comments.

Again, thank you for your consideration.

Yours faithfully,

Hongbing Tan

School of Earth Sciences and Engineering

Hohai University

Nanjing 210098, PR China

Dear Prof. Michael K. Stewart (referee #1),

We are grateful to you for reviewing our work and giving it a fair appraisal. Your comments mean a great deal in improving the manuscript's quality. We have carefully considered all the comments. With help from your comments, we have revised and linguistically embellished the entire text substantially in the revised version. We believe that the revised version will be goal-focused, clearly structured, and linguistically fluent. What follows is our point-by-point response to all your concerns.

**General Comments**

This paper aims to use isotopic measurements in an extremely arid basin in the northeastern Tibetan Plateau to study the water resources of the area and their responses to climate change. The objective is a worthy one, but not very clearly enunciated. The aim of the research is within the scope of HESS and the paper contributes significant new data and analysis of the results. The conclusions reached are substantial and important, particularly as they are for such a climatically difficult area to research.

The extensive data and scientific methods employed allow the authors to reach well-founded interpretations and conclusions. The conclusions reached are substantial. The methods and assumptions are valid and clearly described. The authors give adequate credit to related work and cite appropriate references.

**Reply:** All authors thank you for taking the time to comment on this work. As you can see, a great deal of measurement and data collection has been conducted. We appreciate your positive evaluation for this work. Here are our responses to all of the comments, along with a description of the revisions.

However, the paper is difficult to read and understand. I think this is partly because: 1) it lacks a clear focus and therefore the paper seems to ramble on without direction (as noted above the objective does not seem clearly enunciated), 2) the language is frequently difficult with often very general terms used (such as "Associated Hydrological Indications" in the title") instead of more specific terms that would give a clearer picture, or 3) to a less extent the use of combined words (such as

"spatiotemporal", "spatial-seasonal") makes for difficult reading. Simple words are generally clearer. However, I realise combined words are being increasingly used in scientific papers, and can have specific meanings in some contexts.

**Reply:** Thank you for your kind question about writing skills. We now really realize these questions make readers difficult to well understand the text. Now, a major revision on language, structure and logic for the whole text was done:

1) We have carefully considered each of your concerns and have corrected similar errors throughout the text. These issues have been addressed so that the paper applies to a wider audience. The revised version is linguistically concise, terminologically correct and logically clear. 2) The main topic of this paper should focus on the spatial and temporal variation of isotopes in response to climate changes. Following the revision, we have simplified and highlighted this goal wherever relevant throughout the text, particularly in the Abstract and Introduction. In the Introduction, we simplified the main research objectives into three points. Some sentences or paragraphs that were not closely related to the topic have been deleted. 3) More specific terms and technical terminology were used to express our research findings to adapt them to a wider audience.

A better title might be "Isotopic variations in surface waters and groundwaters of an arid basin and their responses to climate change".

**Reply:** Yes, this is a more reasonable title and we correct title as your idea.

**Specific Comments**

1) The captions to all of the figures need to be more informative. Eg What data does the inset in Fig. 3a show? Also, the light lines in Figs. 4 and 5 show distribution patterns in the δ18O data, but there is no mention of these in the captions. Fig. 11 caption does not identify what is in Figs. 11a and 11b.

**Reply:** Based on your suggestions, we revised all the captions of the figures so that they provide much more clear information for readers. The revisions include data

sources, explanations of special marks, etc.

2) I don't think Fig. 2 is necessary for the paper, although it is interesting.

**Reply:** Agreement. The initial version of Figure 2 provided very limited information. Therefore, we replaced it with a hydrogeological map. In the new Figure 2, hydrogeological information, as well as macroscopic tectonic features, are clearly presented.

3) How is skew defined? The formula used should be explained in Sampling and methods (Sect. 3.) if skew is in fact an important part of the paper. I'm not sure that it is. For example: L258 The paper states "The basin surface water mean $\delta^{18}$O and $\delta$D values were positively skewed by $-0.08$‰ to $1.08$‰ and $0.63$‰ to $10.58$‰, respectively, in the wet seasons." How is this different from "The basin surface water means $\delta^{18}$O and $\delta$D values were more positive by $-0.08$‰ to $1.08$‰ and $0.63$‰ to $10.58$‰, respectively, in the wet seasons."? The latter sentence is much easier to understand (assuming it correctly expresses the intended meaning. If it doesn't, I don't know what the original sentence means.)

**Reply:** Thank you for your suggestion, and we are aware of the use of an ambiguous term in the text. We revised the isotopic signatures throughout the text, especially concerning their spatial and seasonal variations. In the revised version, we defined the spatial and seasonal variation of isotopes in terms of increased and decreased as well as positive and negative.

Again, L614-615 states "The mean values of surface water $\delta18$O and $\delta$D … are negatively skewed from west to east …". Does this mean that "The mean values of surface water $\delta18$O and $\delta$D … decrease from west to east …"?

**Reply:** The sentence has been revised according to your suggestion.

**Technical Comments**

L19 Use of the word "hydrothermal" is not helpful to the reader. Hydrothermal usually refers to water heated by the Earth's internal heat underground, which is not what the

writer means.

**Reply:** Thank you. The hydrothermal conditions are ambiguous. We wanted to express the temperature and precipitation regimes. This term was modified in the revised version.

L137 "Altun fault" not "Alun fault"

**Reply:** Corrected.

L162 "phreatic" not "phrenic"

**Reply:** Corrected.

L178 Should be "the precision was improved to less than ± 0.8 TU" not "more than ± 0.8 TU".

**Reply:** Corrected.

L499 Should be "fell" not "fall"

**Reply:** Corrected.

Dear anonymous referee #2,

We are grateful to you for reviewing our work and giving it a fair appraisal. Your comments mean a great deal in improving the manuscript's quality. We have carefully considered all the comments. What follows is our point-by-point response to all your concerns.

**General comments**

This paper aims to reveal the response of hydrological processes to climate change in an extremely arid basin located on the north-eastern Tibetan Plateau, where it is really an ideal area to evaluate hydrological response to climate changes. With the support of large datasets including stable isotopes and water chemical compositions for a whole basin and its around mountains from different spatial-temporal scales, the manuscript has clarified the entire water cycle processes as well as the isotope hydrology responses to rapid climate changes in an extremely arid basin in the Tibetan Plateau. In addition, the manuscript has also concluded a perspective on the variation trends of water resources under the condition of multiple water sources recharge and climate warming. As my assessment, the topic about the impact of climate change on water resources talked in this manuscript is a general global focused issue. The research objectives, methodology, content and scientific hypotheses of this manuscript are well coincided within the scope of HESS. Thus, I suggest to accept this manuscript, after somewhat improvements. My main concerns are as follows:

1. The title needs to be reconsidered. A title with a clear direction would help to highlight the topic. I believe that the authors may want to emphasize the response of surface water and groundwater isotope variations to climate change. In addition, some of the sub-headings in the text should be also reconsidered so that they can outline the content of the text.

**Reply:** We completely agree with your suggestion. Revealing the isotopic variations in surface waters and groundwaters in response to climate change is the key objective of this paper. According to you and other reviewers' comments, we have revised the title

into *Isotopic variations in surface waters and groundwaters of an extremely arid basin and their responses to climate change*. In addition, we have also reformulated some sub-headings in the Discussion to highlight the topic during the revision.

2. It seems to me that the abstract and introduction need to be furtherly condensed. Although the article is now look -like systematic, a more focused topic is the key to attract the reader's attention.

**Reply:** We emphasized the themes in the Abstract and streamlined the objectives in the Introduction to three points. In the Introduction, we simplified the main research objectives into the following 3 points: 1) elucidating the distribution pattern of surface water and groundwater isotopes in this alpine arid basin at various spatial and seasonal scales; 2) identing and quantifying the main components of the regional water cycle, their timing and spatial heterogeneity; and 3) revealing isotopic hydrological responses to climate change and predicting the trend in the changes of Qaidam Basin water resources. Additionally, the contents of the revised version of the text are all in service of this objective. In the Discussion, some contents were deleted in order to focus on the main topic.

3. As for the text structure, it seems to be a bit confused and may be necessary adjusted. For example, the Section 4.1 should to be better to move after Sections 4.2 and 4.3. It seems to be too long for section 5.1, some subheadings may be added in order to help to fill the gap for readers. Section 5.3 is mainly focused on the groundwater circulation mechanism, which in my view, it should be moved to section 5.2. In addition to the overall structure, the content between paragraphs also needs to be sorted out. Example, the second paragraph of section 5.4 is too long and includes too many ideas. It should be rewritten to be more in point or logical.

**Reply:** Thank you for the good suggestions. Comprehensively consider three reviewers' comments, the structure of the primary text seems to be really disordered and non-logistic. So, in the revised version, the text structure was improved a lot. For example, section 4.1 was streamlined and reorganized as 4.3; section 5.1 was reordered with

paragraphs and subheadings to make it logical; section 5.3 was reorganized as 5.2.4; and section 5.4 was rewritten in 4 points. The revised text is now much more logically and clear for readers. The reformatted framework is as follows:

4.1 Spatial and seasonal characteristics of surface water $\delta^{18}O$-$\delta D$

4.2 Spatial and seasonal characteristics of groundwater $\delta^{18}O$-$\delta D$

4.3 Isotopic variations in different water bodies

5.1 Water cycle information indicated by surface water isotopes

    5.1.1 Atmospheric moisture transport pattern

    5.1.2 Isotopic records of surface water to precipitation

    5.1.3 Climate impact on isotopic spatial and temporal variation

4. Some Figures need to be rehearsed, e.g., regarding Figure 2, a hydrogeological map may be more appropriate for the topic of the paper rather than a simplified tectonic map. In addition, detail information needs to be added in some figure captions.

**Reply:** We proofread all the figures in the full text. Figure 2 was replaced with a hydrogeological map. In addition, missing information in the captions of the remaining figures was added to make them clear.

**Some minor shortcomings please consider to improve:**

Lines 13-14: Delete "The surface water heavy isotopes enrich during the wet season and deplete during the dry season". The abstract should highlight key findings.

**Reply:** Deleted.

Line 19: hydrothermal conditions are not an appropriate terminology, I guess the author wants to express temperature and precipitation regimes.

**Reply:** Thank you for your revision. temperature and precipitation regimes do better match our meaning. All ambiguities about it have been corrected.

Line 21: Delete "ice/snow".

**Reply:** Deleted.

Line 28: Keywords should be ordered according to major and second categories and then followed specific research directions

**Reply:** The new Keywords are as follows: Water cycle; climate change; Isotope hydrology; spatial and temporal variation; Qaidam Basin.

Line 37: Delete "region of the".

**Reply:** Deleted.

Line 57: "These issues require an in-depth investigation" is a repetition of the "several questions remain to be resolved" in Line 55.

**Reply:** "These issues require an in-depth investigation" was deleted.

Lines 91-95: No more than 3 goals of the study were considered reasonable. The authors were advised to streamline the study objectives. The order of the research objectives should also be expressed more logical.

**Reply:** The revised objectives are: 1) elucidating the distribution pattern of surface–groundwater isotopes and the composition changes of water sources in alpine arid basin at various spatial and seasonal scales; 2) tracing the entire water cycle process around the mountain–basin watersheds in the Qaidam Basin; and 3) revealing isotopic hydrological responses to climate change and predicting the trend in the changes of Qaidam Basin water resources.

Line 118: "Hydrogeology and structure" can be changed to Basic hydrogeological setting

**Reply:** Modified as suggested.

Line 137: Altun is not Alun

**Reply:** Corrected.

Line 162: phreatic groundwater is not phrenic groundwater

**Reply:** Corrected.

Line 206: Section 4.1 seems to duplicate 4.2 and 4.3 in terms of structure, although the contents are different. I think the contents of 4.1 can be moved after 4.2 and 4.3 after simplification.

**Reply:** According to your suggestion, section 4.1 was simplified and adjusted to 4.3.

Line 245: Specific terminology in the whole text should be used, e.g."positively skewed" is not a terminology and it easily causes ambiguity. For isotopic expression, please use "depleted or enriched" to decipher their variations.

**Reply:** We apologize for the ambiguity caused by lay terms. The terms used to define isotopic changes throughout the text have been revised following your suggestion.

Line 254: "seasonal and spatial variations" should be replaced by "spatial and seasonal variations".

**Reply:** Corrected.

Lines 263-268: The description of spatial variations should be placed before the description of temporal variations in order to match the title.

**Reply:** Corresponding sentences have been reordered in this chapter.

Line 292: In chapter 5.1, some subheadings need to be added. Also, the order of the paragraphs should be logically adjusted. The paragraph on moisture tracing should be placed previously, and the content of Lines 309-330 should be swapped with Lines 356-376.

**Reply:** Thank you for your suggestion. We reset the paragraph order and assigned subheadings to the subsections as previous statement.

Line 377: please revise it as Line 254.

**Reply:** Revised.

Line 378: According to my understanding, regional warming and humidification trends is better than the extent of warmth and humidity in the region

**Reply:** Agree. it was modified.

Line 386: The headings and subheadings of chapter 5.2 also need to be reconsidered to summarize the topic of the paragraph. Additionally, in this chapter, the main idea of the article should be strengthened but some discussions that are not closely related to the scientific issues should be shorten or deleted.

**Reply:** We have rewritten the title and sub-title of chapter 5.2 and revised the previous chapter 5.3 to 5.2.4, as it also deals with groundwater circulation mechanisms. In addition, the expansions that were not closely related to the scientific issues were removed.

Line 394: 5.2.1 In the extremely arid region, recharge from precipitation occurs mainly during the wet season, which is a general idea and need not to be emphasized.

**Reply:** The title of 5.2.1 has been streamlined.

Line 445: Can the authors please confirm that it is different seasons that is correct? I think the author meant different phases.

**Reply:** The "different phases" is right.

Lines 502-504: These sentences can be deleted for repetition in the result section.

**Reply:** Deleted.

Lines 651-562: This is a repetition of the previous sentence.

**Reply:** Deleted.

Line 599: Figure 10b is redundant, although it presents a lot of data, as Figure 10a is sufficient to show the striking climate change trends in the study area. It could also be removed from the text, as the obvious redundancy of information is not advocated.

**Reply:** Agreed. The revised version retains only Figure 10a.

Line 604: The authors should report the source of the data in Figure 11. Because some of the data are obviously collected from the literature. I would suggest putting figure

11 in the annex as the highlights of this section are already shown in Figure 12.

**Reply:** The references involved in Figure 11 were added. We have considered your suggestion and endorse it. However, Figure 11 was retained in the text, although the partial information within it partially overlaps with Figure 12. This is because it visualizes the inter-annual variability of the isotope signals.

Again, thank you for your recognition of this work. In conjunction with your comments, we have revised and linguistically embellished the entire text substantially in the revised version. So, the revised version is very different from the initial, both concerning scientific and data presentation and the English language or grammar. We believe that the revised version will be goal-focused, clearly structured, and linguistically fluent.

Dear anonymous referee #3,

We are grateful to you for reviewing our manuscript and objectively comments. Your comments encourage us to greatly improve the quality of manuscript. We have carefully considered all the comments. What follows are our point-by-point response to all your concerns.

This paper combines a very rich dataset of water isotopologues to provide conceptual models of the continental water cycle components in the different parts of the extensive, arid Qaidam basin in the Tibetan Plateau. The authors reach interesting conclusions on constrasting functioning of the sources and fates of water pathways and surface-groundwater exchanges in different part of the study area, with important implications for ongoing and future trajectories in a changing climate.

While this could be a significant contribution to the community and the topic fit wells in HESS readership, in my view the manuscript suffers from an inadapted structure that makes it hard to follow, and to fully assess the soundness of the proposed analyses and discussions. Due to this and other issues, I recommend a very substantial rewriting effort before this manuscript can be considered for publication. This would include a further effort on the level of written English as well; I tried to list some items in the technical comments, but this is not the role of a reviewer and a complete proof-reading, if possible, by a native speaker, will be needed.

**General comments**

The current structure of paper seems to be the biggest issue at this stage. On this matter I concur with the other comments, who notably summarized the main issues: lack of clear focus, difficult vocabulary and formulations (in the main text and some titles).

In particular, it seems the Discussion section is too long and essentially mixes a lot of additional results and discussion. I strongly advise the authors to rework this section in connexion with the Results section. Given the number of datasets and related analysis presented, special care should be given in crafting a more precise focus and red thread throughout these sections, in connexion with more clearly stated objectives and

research questions.

**Reply:** We apologize for the confusion caused by the disorganized structure and sloppy language. We have therefore summarized your and two other referees' relevant comments and made a major rewriting effort throughout the text. These enhancements focus on the study objectives and discussion chapters. Specific efforts included 1) the direction of the article's focus was clarified, 2) the logical framework was reorganized, 3) discussions of insignificant relevance to the main results were removed, and 4) the language and terminology were polished.

In the Introduction, we simplified the main research objectives into the following 3 points: 1) elucidating the distribution pattern of surface water and groundwater isotopes in this alpine arid basin at various spatial and seasonal scales; 2) identing and quantifying the main components of the regional water cycle, their timing and spatial heterogeneity; and 3) revealing isotopic hydrological responses to climate change and predicting the trend in the changes of Qaidam Basin water resources.

Moreover, section 4.1 was streamlined and reorganized as 4.3; section 5.1 was reordered with paragraphs and subheadings to make it logical; section 5.3 was reorganized as 5.2.4; and section 5.4 was rewritten in 4 points. The revised text based on your comments is logically clear and progressive. The reformatted framework is as follows:

4.1 Spatial and seasonal characteristics of surface water $\delta^{18}$O-$\delta$D

4.2 Spatial and seasonal characteristics of groundwater $\delta^{18}$O-$\delta$D

4.3 Isotopic variations in different water bodies

5.1 Water cycle information indicated by surface water isotopes

   5.1.1 Atmospheric moisture transport pattern

   5.1.2 Isotopic records of surface water to precipitation

   5.1.3 Climate impact on isotopic spatial and temporal variation

The embellishment of the above headings is based on summarizing the content of the paragraph and highlighting the main subject of the whole article, to strengthen the logic and readability of the article.

**Specific comments**

L20: "caused by climate warming" during which period?

**Reply:** It refers to current warming. Corrected.

L22: what are "structural channels"?

**Reply:** "structural channels" is ambiguous, and what we are trying to convey is that the favorable water conduits, such as faults and fissures.

L29-30: I disagree that an "in-depth study of the hydrological cycle is a prerequisite of water management"; Effective water management practice implementation may be hindered not by lack of knowledge but also geopolitical reasons, although it is a case-by-case matter. I understand the authors need to justify the signficance of their study, so I'd recommend to be more balanced, for example rephrasing as follows "an in-depth study of the hydrological cycle processes is a prerequisite for accurate trend forecasting, and helps to design efficient water resource management strategies."

**Reply:** Thanks for the suggestion to modify it accordingly. Corrected.

L62: H & O are not isotopes per sé. May an appropriate formulation would be "The isotopes of hydrogen and oxygen elements are useful tracers of the water cycle..."

**Reply:** Corrected.

L66-68: why talking only about stable isotopes (excluding 3H), and then mentioning 3H in the next sentence? Both stable and non-stable types are useful tracers in the applications cited.

**Reply:** These two have been modified. Both stable and non-stable isotopes are useful tracers in the text.

L70-71: an additional reference using stable istopes and tritium may be Rodriguez et al.

(2019) and a recent paper on the topic (Benettin et al., 2022)

**Reply:** The most recent relevant literature on stable isotopes and tritium tracing of the water cycle like your issued has been cited in the corresponding position.

L81: I'm not sure to understand how tectonic patterns are caused by "hydrological, climatic and hydrogeological conditions". Isn't the other way around?

**Reply:** You're right. It was modified.

L91-95: objectives #1 to #2 seem somewhat vague and overlap. Also, perhaps the authors should consider having more focus objective than #3, e.g. instead of tracking "the entire water cycle" --> "identying and quantifying the main components of the regional cycle, their timing and spatial heterogeneity" might be already quite ambitious and enough?

**Reply:** The objectives of the article in the Introduction have been simplified at your suggestion and that of other Referees to 3 points. The aims are 1) elucidating the distribution pattern of surface water and groundwater isotopes in this alpine arid basin at various spatial and seasonal scales; 2) identing and quantifying the main components of the regional cycle, their timing and spatial heterogeneity; and 3) revealing isotopic hydrological responses to climate change and predicting the trend in the changes of Qaidam Basin water resources.

L96-100: This part is essentially a repetition of the above, please consider removing it.

**Reply:** Yes. This part was removed.

L159: is the sampling frequency or sampling period?

**Reply:** It is the sampling period.

L163-164: Providing a quick idea of the rain and snow(melt) sampling frequency would be useful.

**Reply:** Relevant information has been added.

L230: I guess mentioning "evaporative fractionation" instead of "evaporation" helps for

the analysis the plotting below the LMWL

**Reply:** You're right. It was modified.

L253: In this section, having a dual-isotopes plots per locations (possibly in the supplmentary materials) would help the analysis; it seems this the current Fig. 7, but why does the latter comes much latter and is being only quickly mentioned in the main text (L357, L393) ?

**Reply:** We apologize for the confusion caused by inconsistent content. A new Figure corresponding to the characterization of the results section has been uploaded and its naming has been redefined from Figure 3 to Figure 5 in the text.

L285-288: I am not convinced that this assertion is supported by the data. Can the authors further develop and articulate this hyptohesis?

**Reply:** It is really seemed to be very abrupt here. This has been rephrased and moved to be previous relevant expressions.

L337-340: This is interesting and should probably come earlier in the analysis, at least before general statements as those found e.g. L328-330?

**Reply:** Agreed. Under the new framework, it has been adjusted to the appropriate prominence.

L351-352: Is 3 permill (from 10 to 7) a substantial difference in d-excess?

**Reply:** Yes, this is a difference in mean d-values. Please refer to Table S1 for relevant data. Consistent with our results, previous works have shown that there is a substantial difference in the d value of water bodies originating on the north and south sides of the Tanggula Mountains in terms of their characteristics. This is caused by differences in moisture sources.

L400-519: In line with the general comments, it seems quite strange that the information content and results of tritium analysis only appears so late in the paper.

**Reply:** We agree with you and have resolved the confusion. The information content

and results of tritium analysis in surface water and groundwater have been added to the Results section.

L499: Typical tritium detection limit is around 0.05 TU or less, so this gives considerable range from "<3TU". Could the authors explain why they grouped all samples below 3TU and those "below detection limit"? And also provide their detection limit somewhere in the methods?

**Reply:** Yes, as you say, "less than 3 TU" and "below detection limit" at the same time are ambiguous. We are trying to convey low tritium concentration here. The ambiguity in the text was modified.

L578: what do the authors mean by "the cyclical nature of climate change"? The long-term glacial-interglacial dynamics? Please rephrase, as it reads it may be understood that current global warming is cyclical (against all evidence synthetized by the IPCC).

**Reply:** Rephrased.

L587-589: I did not understand how the authors can predict a large-scale decrease after the increase in water resources, please clarify.

**Reply:** This issue has not been well clarified due to confusing logic. In the revised version, some literature data was added to support this conclusion. We rewrite this paragraph in order to convince it to be accepted by readers.

L592-593: does this involve regional evaporation recycling mechanisms? AS it stands, little of this is directly discuss in the paper, yet it seems like an important piece of the puzzle.

**Reply:** The evaporation recycling mechanism is actually existent here, especially in these watersheds where precipitation is sparser. Corresponding mechanisms are added here to couple other hydrometeorological elements. We would like to state that in this extremely arid basin, intense evaporation is one of the most dominant climatic features. Therefore, this premise is present by default, although there seem to be regional differences between the watersheds. Limited by the paper's length, we did not develop

a detailed discussion of evaporation recycling mechanism. It is clear that these proposals point the way we forward. The modifications in the text are as follows:

3) In the central basin (Nomhon, Golmud, and Yuka Rivers), there is long-term large-scale groundwater mining during the agriculture and industry development, accompanied by strong local evaporation. The sparse precipitation in the source area led to a melt dependence, although the surface water and groundwater recharge here is relatively stable.

L637: similar comment as for L578 ("cyclical nature of climate change")

**Reply:** Rephrased.

**Technical comments**

L8: "climate warming" reads odd to me. maybe "climate change" or "global warming" instead? It is found several times in the manuscript

**Reply:** These ambiguities have been modified in the whole text.

L13: heavy isotopes do not enrich, more correctly stated "the water is enriched in heavy isotopes"

**Reply:** Modified.

L18: "regulated" reads strange; "controlled", maybe?

**Reply:** Modified.

L22: what is the "impending climate change process"? Perhaps "In the face of ongoing environmental changes, ..."

**Reply:** Modified.

L47: replace "and a substantial steep...globally" by "with a steep rise in temperature overall".

**Reply:** Modified.

L49: "0.53°C per decade"?

**Reply:** Yes, you are right. It was corrected as suggest.

L77: please define a "confined regional unit scale" or rephrase.

**Reply:** Rephrased.

L84: "to study the entire process of the water cycle" --> "to get a comprehensive view of the water cycle"?

**Reply:** Modified.

L85: "isotopic composition" rather than "isotopes"

**Reply:** Modified.

L92: "in this alpine arid basin"

**Reply:** Modified.

L131: "extensive" instead of "expansive"?

Reply: Modified.

L162: I am not sure what "phrenic" means here; is it "phreatic"?

**Reply:** Modified.

L245: by "positively skewed" I guess the authors mean that it is more enriched; I find the latter expression more correct and self-explanatory.

**Reply:** Terminology on isotope changes throughout the text has been checked and modified

L29 - Fig. 3: the figure is very small and symbols are hard to tease out; consider using a larger, vertically stacked figure (with higher DPI / less compression as well, it is quite pixelized at present)

**Reply:** An updated figure consistent with the main text has been uploaded and now it looks clear a lot.

L313: what do the authors mean by "an augmentation in the continental characteristics

of water vapor"? Please clarify and/or rephrase.

**Reply:** Rephrased. Again, we apologize for the confusion.

L457 - Fig. 8: The scale is hard to read; I suggest the authors remove the color from the background (with a simplified map) and use a color scale to isotopic ratios and 3H. I would also suggest to duplicate the panels to clearly separate surface and groundwater, as surface is here only shown for tritium.

**Reply:** Thank you for your suggestions, we have modified Figure 8 based on them, the new graphic is more aesthetically pleasing and helps to directly present our findings.

References

Benettin, P., Rodriguez, N. B., Sprenger, M., Kim, M., Klaus, J., Harman, C. J., ... & McDonnell, J. J. (2022). Transit time estimation in catchments: Recent developments and future directions. Water Resources Research, 58(11), e2022WR033096.

Rodriguez, N. B., Pfister, L., Zehe, E., & Klaus, J. (2021). A comparison of catchment travel times and storage deduced from deuterium and tritium tracers using StorAge Selection functions. Hydrology and Earth System Sciences, 25(1), 401-428.

---

## Author Response (AR2)

Dear Prof. Markus Hrachowitz,

Thank you for your time and attention in handling our manuscript, as well as for providing us an opportunity to revise and improve its quality. We are pleased to submit a revised R2 version. The manuscript has been moderately revised according to Prof. Michael K. Stewart's comments and suggestions. In addition, the reference list of the manuscript has been compiled according to HESS standards. We believe that the revised R2 version is now significantly improved from the R1 version with respect to the English language or grammas. We would like to sincerely thank you, two reviewers for these valuable comments. The following are point-by-point responses to the reviewers' comments.

Again, thank you for your consideration.

Yours faithfully,

Hongbing Tan

School of Earth Sciences and Engineering

Hohai University

Nanjing 210098, PR China

General Comment

This paper has been revised after several reviews. The paper is now much more readable and I recommend that the paper now be accepted for publication in HESS, with consideration of the detailed comments below.

Detailed Comments

Minor details noted while reading the paper:

Page 1, Line 16: Suggest change to "..The H-O isotopic compositions of rivers in .." from "..The spatial H-O isotopic compositions in .."

**Reply:** Corrected.

P10 & 11, Fig. 3 and Fig. 4 captions: Should be "The dashed lines .." not "The light lines .."

**Reply:** Corrected.

P15, L315: Should be ".. and increase river flows." not ".. and rapidly recharge the river." (Note, groundwater systems are recharged, while river flows are increased.)

**Reply:** Corrected.

P16, L345: Should be ".. and groundwater feed the river water during the wet season .." not ".. and groundwater recharge the river water during the wet season .."

**Reply:** Corrected.

P19, L370: Should this be Fig. 4 not Fig. 5?

**Reply:** Corrected.

P20, L 402: Should be ".. and further depleted the groundwater .." not ".. and was further depleted in the groundwater .."

**Reply:** Corrected.

P20, L412: Should be ".. is the combination of different periods of atmospheric .." not

".. is different periods atmospheric .."

**Reply:** Corrected.

P20, L 413: Should be ".. decreasing .." not ".. decreased .."

**Reply:** Corrected.

P23, L459: Should be ".. owing to smaller and steadier meltwater recharge .." not ".. owing to less and more steady meltwater recharge .."

**Reply:** Corrected.

P24, L491: ".. isotopic measurements in water bodies over the past 40 years suggests that there is a range of interannual variability .." not ".. isotopic variability in water bodies over the past 40 years suggests that there is a variable degree of interannual variability .."

**Reply:** Corrected.

P25, L513: ".. storage increases and lake expansions. The .." not ".. storage increase and lake expansion. The .."

**Reply:** Corrected.

P30, L575: ".. River base flow is maintained by groundwater discharge during the dry season, and rivers receive varying proportions .." not    ".. The base flow is maintained by groundwater recharge during dry season, while receiving varying proportions .."

**Reply:** Corrected.

P30, L593: ".. accelerating the water cycle and increasing water storage in the .." not ".. accelerating water cycle and raising water storage uncertainties in the .."

**Reply:** Corrected.

P31, L597: ".. areas becomes severely out of balance due to extreme climatic changes." not ".. areas becoming severely out of balance undergone climatic extreme changes."

**Reply:** Corrected.